# On the Difficulty of Learning a Meta-network for Training Data Selection

Zilin Du [1]  Junqi Zhao [1]  Boyang Albert Li [1]

## Abstract

Synthetic data are increasingly used to train neural networks, yet distributional mismatch with real data limits their effectiveness when used indiscriminately. A common strategy is to learn data weights via bi-level optimization, which we refer to as Meta-learning for Training-data Selection (MTS). Interestingly, in practice, MTS often performs below expectation. We identify two obstacles in properly training MTS: a poor gradient signal-to-noise ratio (GSNR), which causes optimization difficulties, and lack of informative features that correlates with data quality. We present a mathematical analysis of MTS, which reveals the dynamics of normalized data weights and the relation between disparate data quality and poor GSNR. The analysis suggests a a simple yet effective solution: increasing the batch size. Further, we propose a set of informative features that capture the positions of training data in their distributions and training dynamics. Experiments across four benchmarks show consistent improvements, achieving average gains of 5.49% over training without selection and 2.89% over the strongest baseline.[1]

## 1. Introduction

It is often desirable to selectively train on a subset of available data, especially when the training and test data are not identically distributed (Lee et al., 2024; Kang et al., 2024; Liu et al., 2024; Xie et al., 2023). A prominent use case scenario these days is training with synthetic data (Fan et al., 2024; Gala & Xie, 2024; Li & Li, 2025), which has become increasingly practical as generative models improve. Nevertheless, synthetic data often exhibit distributional shifts or artifacts that distinguish them from real data (Hataya et al., 2023; He et al., 2023; Liu et al., 2025).

A popular approach for training data selection is to assign soft weights $w_i$ to each training sample $(x_i, y_i)$ (Ren et al., 2018; Shu et al., 2019) in order to select effective training data or align the training and validation/test distributions. We define $w_i := s_\phi(x_i)$, where $s_\phi$ is a data selection network parameterized by $\phi$. The main network $f_\theta(x)$ is parameterized by $\theta$. With training data points $(x_i, y_i)$, validation data $(x_i^{\text{v}}, y_i^{\text{v}})$, and per-instance loss $\ell(\cdot)$, the mini-batch training loss and validation loss are defined as

$$\mathcal{L}_{\text{train}}(\theta, \phi) := \Big( \sum_{i=1}^{N} w_i \Big)^{-1} \sum_{i=1}^{N} w_i \ell(\theta, x_i, y_i), \quad (1)$$

$$\mathcal{L}_{\text{val}}(\theta) := N^{-1} \sum_{i=1}^{N} \ell(\theta, x_i^{\text{v}}, y_i^{\text{v}}), \quad (2)$$

where $N$ denotes the batch size. To optimize $\phi$, prior works adopt a one-step approximation of the complex bi-level optimization. At iteration $t$:

$$\phi_{t+1} = \phi_t - \eta_\phi \nabla_{\phi_t} \mathcal{L}_{\text{val}}(\theta_t - \eta_\theta \nabla_{\theta_t} \mathcal{L}_{\text{train}}(\theta_t, \phi_t)) \quad (3)$$

and $\eta_\theta$ and $\eta_\phi$ are the learning rates. We refer to this framework as Meta-learning for Training-data Selection (MTS).

Despite its intuitive appeal, MTS often performs under expectation. In Table 2, we show that naive training of MTS with raw image input to $s_\phi(\cdot)$ and small batches fails to outperform training without data selection. What causes performance issues of MTS?

We identify two obstacles in MTS. First, the gradient signal-to-noise ratio (GSNR) of the selection network is exceedingly low (Figure 1). Though moderate levels of variance in the stochastic gradients can serve as implicit regularization (Smith et al., 2020; Mignacco & Urbani, 2022; Zhou et al., 2020), excessive variance may overwhelm the gradient signal and harm optimization and generalization (Sankararaman et al., 2020; Liu et al., 2020).

We present a comprehensive analysis of the GSNR issue, which shows that the MTS training tends to suppress most data weights $w_i$, which in turn leads to high sensitivity of the gradient update to $\phi$ to the highest data weight. This

[1] College of Computing and Data Science, Nanyang Technological University, Singapore. Correspondence to: Boyang Albert Li <boyang.li@ntu.edu.sg>.

Proceedings of the 43$^{rd}$ International Conference on Machine Learning, Seoul, South Korea. PMLR 306, 2026. Copyright 2026 by the author(s).

[1] Our code is available at https://github.com/ZILIN003/MTS.

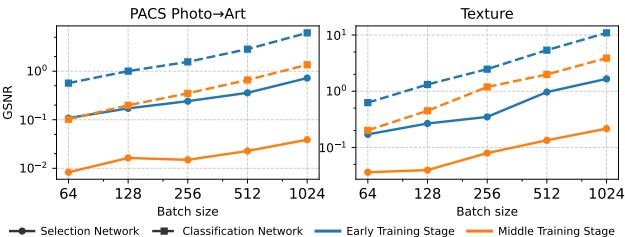

*Figure 1.* The gradient signal-to-noise ratio (GSNR) of the selection network is lower than that of the ResNet classification network by roughly one order of magnitude. However, increasing the batch size can help. Early and middle training stages correspond to 20% and 60% of the total training steps. Results are computed as averages over 100 batches.

directly results in low GSNR. The analysis also presents a simple remedy: increasing the batch size.

The second obstacle is that input features to the selection network do not correlate well with data effectiveness in training. Even if the network can be trained to convergence due to a good GSNR, it may still lack the information to distinguish between high-quality and low-quality data. We propose a set of informative features, representing the positions of training data in the training and validation distributions and their interaction with the training process. These distributional and dynamic features may be difficult to learn from the raw input $x_i$ alone, as they may involve relations among multiple data points and the training process.

Good GSNR and high-quality features are both necessary for effective training of the selection network and complementary to each other. Intuitively, high-quality features ensure good solutions exist in the hypothesis space, or the set of all functions representable by the selection network; decent GSNR allows us to find such good solutions quickly. Empirically on the task of synthetic data selection, we demonstrate substantial enhancement in MTS performance when the two solutions are combined, but weak or no improvements when they are employed in isolation. On 4 different datasets and 15 different settings, our method attains strong performance, surpassing the baseline without data selection by 5.49% and the strongest baseline by 2.89%.

Our contributions are summarized as follows:

- We provide a mathematical analysis revealing a fundamental obstacle that hinder effective training of MTS: a low gradient signal-to-noise ratio (GSNR) in the selection network. The theoretical analysis identifies the cause of low GSNR and a simple and effective remedy: increasing the batch size.

- We design informative features that correlate with the effectiveness of training data and allow the data selection network to function as intended.

## 2. Related Work

### 2.1. Synthetic Images for Image Classification

Recent advances in diffusion models have made synthetic images a practical resource for training deep models (Feng et al., 2024; Xie et al., 2024; Du et al., 2023), and this capability has been widely leveraged in image classification (Bulent Sariyildiz et al., 2022; Jiang et al., 2025; Singh et al., 2024; Tian et al., 2024). However, synthetic images often exhibit distributional shifts and artifacts relative to real data, which can degrade generalization if used indiscriminately (Hataya et al., 2023; He et al., 2023). To mitigate these issues, prior work has largely focused on generation-side improvements, including fine-tuning generative models toward the target distribution (Kim et al., 2024; Azizi et al., 2023; Wang et al., 2024b), conditioning on real images (Islam et al., 2024; Liu et al., 2025; Trabucco et al., 2023), and prompt engineering strategies (Yuan et al., 2024; Shipard et al., 2023; Dunlap et al., 2023; Zhao et al., 2024).

### 2.2. Weighing Training Data

The notion of assigning distinct weights to data points in the training loss finds applications in diverse problems such as data pruning and domain adaptation. Data pruning aims to find a subset of training data that are as effective as the whole training set (Toneva et al., 2019; Paul et al., 2021; Pleiss et al., 2020; Feldman et al., 2020; Xia et al., 2022; Zheng et al., 2023). This is equivalent to binary 0/1 weights. In domain adaptation, the data weights may help align the training data distribution with the validation distribution (Wang et al., 2024a; Sugiyama et al., 2007; Xie et al., 2023; Bickel et al., 2009; Choi et al., 2021), often with continuous weights, rather than 0/1 weights. A closely related line of work is the selection of synthetic training images, which often employs CLIP similarity (Dunlap et al., 2023; Bansal & Grover, 2023; He et al., 2023; Zhao et al., 2024; Li et al., 2024; Wang et al., 2024b) or auxiliary classifier confidence (Bansal & Grover, 2023; Lampis et al., 2023). Despite their success, the weights are largely assigned from heuristic rules rather than learned from data. The current paper focuses on the selection of synthetic training data, but the technique may potentially be applied to other problem settings. In the next subsection, we review techniques that learn data weights using gradient-based optimization.

### 2.3. Gradient-based Hyperparameter Optimization

Gradient-based hyperparameter optimization (Franceschi et al., 2018; Kwon et al., 2023; Dong et al., 2025; Bao et al., 2021) enables automatic tuning of diverse hyperparameters, such as learning rates (Maclaurin et al., 2015), dropout

rates (Lorraine et al., 2020), architectures (Liu et al., 2018; Zhang et al., 2021), and loss functions (Kim & Hospedales, 2025). Meta-learning for Training-data Selection (MTS) is the learning of data weights under this framework. Ren et al. (2018); Wang et al. (2022) employ a separate hyperparameter for each data point, but estimating a hyperparameter from a single training data point may cause high variance and limit knowledge sharing. To alleviate this issue, Fan et al. (2023) learn one weight hyperparameter for each data domain, so that knowledge can be shared across data in the same domain. Shu et al. (2019), on the other hand, propose to calculate the data weights using a lightweight neural network, which is a generalizable function learned across all training data. However, they only employ a single feature, the training loss, which does not capture rich patterns in data distribution and generalization behavior.

In this work, we provide a theoretical analysis that applies to both the individual data weight setting (Ren et al., 2018; Wang et al., 2022) and the data-weight network setting. Further, we introduce a rich input feature representation for the data-weight network to better characterize the distributional information of training data and their potential for generalization.

### 2.4. Gradient Signal-to-Noise Ratio (GSNR)

Earlier work has shown that stochastic gradient noise can act as an implicit regularizer, improving generalization (Smith et al., 2020; Mignacco & Urbani, 2022) and helping optimization escape sharp minima (Zhou et al., 2020). Beyond analyses of gradient noise, some works have investigated the gradient signal-to-noise ratio (GSNR), though this line of research remains relatively underexplored. Theoretically, Liu et al. (2020) show that a large GSNR is beneficial for the generalization of deep networks, while Rainforth et al. (2018) analyze how GSNR varies for the encoder and decoder of variational autoencoders as the number of importance samples increases. Empirically, GSNR has been used as a heuristic for domain generalization (Michalkiewicz et al., 2023), knowledge distillation (Huang et al., 2025), and neural architecture search (Sun et al., 2023), among others. In this work, we adopt GSNR as a key analytical tool for understanding MTS.

## 3. Difficulty in Training Selection Network

In this section, we formalize the MTS problem and provide a theoretical analysis of the cause of the low signal-to-noise ratio in the selection network gradient.

### 3.1. Notations and Problem Definition

We first introduce the notations. We have a training set $\mathcal{D}_{\text{train}} = \{(x_i, y_i)\}_{i=1}^{|\mathcal{D}_{\text{train}}|}$, where each input $x_i$ is asso-

ciated with a class label $y_i$, and a small validation set $\mathcal{D}_{\text{val}} = \{(x_j^{\text{v}}, y_j^{\text{v}})\}_{j=1}^{|\mathcal{D}_{\text{val}}|}$. We train a classifier network $f_\theta(\cdot)$, parameterized by $\theta$, and a data selection network $s_\phi(\cdot)$, parameterized by $\phi$. The data weight for training sample $i$ is defined as $w_i := s_\phi(x_i) \in (0, \infty)$. Let $\ell(x_i, y_i, \theta)$ denote the per-sample training loss, such as cross-entropy, and define its gradient as $g_i(\theta) := \partial \ell(x_i, y_i, \theta)/\partial\theta$. Similarly, for validation samples, we define $\gamma_i(\theta) := \partial \ell(x_i^{\text{v}}, y_i^{\text{v}}, \theta)/\partial\theta$.

We denote the overall training and validation losses by $\mathcal{L}_{\text{train}}$ and $\mathcal{L}_{\text{val}}$, respectively. The gradient of the selection network output is given by $h_i := \partial s_\phi(x_i)/\partial\phi$. Finally, we define the normalization scalar $S := \sum_{i=1}^{N} w_i$ and the normalized weight vector $p := [p_1, \ldots, p_N]$ where $p_i = w_i/S$.

With data weights $w_i$, the mini-batch training loss is defined as

$$\mathcal{L}_{\text{train}}(\theta, \phi) = \frac{1}{S} \sum_{i=1}^{N} w_i \, \ell(x_i, y_i, \theta). \tag{4}$$

Normalization by $S$ ensures that the effective weight vector $p = [w_i/S]_{i=1}^{N}$ sits on the $(N-1)$-dimensional probability simplex.

The mini-batch validation loss does not use the data weights:

$$\mathcal{L}_{\text{val}}(\theta) = \frac{1}{N} \sum_{i=1}^{N} \ell(x_i^{\text{v}}, y_i^{\text{v}}, \theta). \tag{5}$$

### 3.2. Optimizing the Data Weights

The central issue is to learn the data selection parameters $\phi$ from data. To this end, we adopt the following nested optimization (Shu et al., 2019; Ren et al., 2018):

$$\phi^* = \arg\min_{\phi} \ \mathcal{L}_{\text{val}}(\theta^*)$$
$$\text{s.t. } \theta^* = \arg\min_{\theta} \ \mathcal{L}_{\text{train}}(\theta, \phi) \tag{6}$$

Note that $\theta^*$ does not depend on $\phi$ directly. Rather, its dependence on $\phi$ is through the optimization of $\mathcal{L}_{\text{train}}$ as $\phi$ determines the data weights. We seek $\phi^*$ that produces a $\theta^*$ that generalizes the best to $\mathcal{L}_{\text{val}}$.

Directly optimizing Eq. 6 is intractable as each gradient update of $\phi$ would require solving an expensive optimization problem for $\theta$. Following common practice, we adopt an approximate hypergradient for $\phi$ with a one-step look-ahead update of $\theta$. At iteration $t$, we perform the following updates:

$$\phi_{t+1} = \phi_t - \eta_\phi \frac{d}{d\phi_t} \mathcal{L}_{\text{val}}(\theta_t - \eta_\theta \frac{\partial}{\partial\theta_t} \mathcal{L}_{\text{train}}(\theta_t, \phi_t)), \tag{7}$$

$$\theta_{t+1} = \theta_t - \eta_\theta \frac{\partial}{\partial\theta_t} \mathcal{L}_{\text{train}}(\theta_t, \phi_{t+1}), \tag{8}$$

where $\eta_\theta$ and $\eta_\phi$ are the respective learning rates. Though it is not necessary to perform equal number of updates to $\theta$

and $\phi$, we empirically find equal number of updates to work well.

**Lemma 3.1** (The hypergradient of $\phi$). *The update to $\phi$ in (7) is computed as*

$$\phi_{t+1} = \phi_t - \eta_\theta \eta_\phi \Delta\phi_t, \tag{9}$$

*where the hypergradient update $\Delta\phi_t := \frac{1}{\eta_\theta} \frac{d}{d\phi_t} \mathcal{L}_{val}(\theta_t')$ is (Appendix A.1)*

$$\Delta\phi_t = -\frac{1}{S}\left(\frac{1}{N}\sum_{i=1}^N \gamma_i(\theta_t')\right)^\top \sum_{i=1}^N g_i(\theta_t)\left(h_i(\phi_t)^\top \right.$$
$$\left. -\frac{w_i}{S}\sum_{j=1}^N h_j(\phi_t)^\top\right), \tag{10}$$

*where $\theta_t'$ is $\theta$ after the one-step update:*

$$\theta_t' = \theta_t - \frac{\eta_\theta}{S}\sum_{i=1}^N w_i g_i(\theta_t), \tag{11}$$

*and $\eta_\theta$ and $\eta_\phi$ are learning rates.*

### 3.3. Dynamics of the Data Weights

We first analyze dynamics of $p_i = w_i/S$ in the simplified setting where each $w_i$ is a separate hyperparameter, rather than the output of a network. This is basically the setting of Ren et al. (2018); Wang et al. (2022). In the complete setting, $w_i$ is the output of network $s_\phi(\cdot)$ and updated through parameters $\phi$. However, the update directions on $w_i$ with and without $\phi$ are similar.

The hypergradient of $\mathcal{L}_{val}$ against $w_i$ is (Appendix A.2),

$$\frac{d\mathcal{L}_{val}}{dw_i} = -\eta_\theta \frac{1}{S}\bar{\gamma}(\theta_t')^\top \left[g_i(\theta_t) - \sum_k p_k g_k(\theta_t)\right], \tag{12}$$

where $\bar{\gamma}(\theta_t') = \sum_i^N \gamma_i(\theta_t')$. For simplicity, we drop the dependence on $\theta_t$ and define scalars

$$a_i := \bar{\gamma}^\top g_i, \quad \bar{a} := \sum_k p_k a_k. \tag{13}$$

We can consider $a_i$ as a measure of the alignment between the validation gradient $\gamma$, estimated over the entire validation batch, and the gradient of the $i$-th training data point $g_i$. In comparison, $\bar{a}$ measures the overall alignment weight-averaged over the training batch. Treating $w_i$ as separate hyperparameters, the gradient descent update to $w_i$ is

$$w_i \leftarrow w_i - \eta_w \frac{d\mathcal{L}_{val}}{dw_i} = w_i + \frac{\eta_\theta \eta_w}{S}\left(a_i - \bar{a}\right), \tag{14}$$

where $\eta_w > 0$ is the learning rate. Therefore, $w_i$ will increase if $a_i > \bar{a}$ and decrease if $a_i < \bar{a}$. In other words,

examples whose gradient $g_i$ is more aligned with the validation gradient $\bar{\gamma}$ than the current weight-averaged alignment $\bar{a}$ are up-weighted, while examples whose alignment is weaker than $\bar{a}$ are down-weighted. Overall, this steers the training gradient toward the validation gradient direction.

However, this also introduces a competitive, winner-takes-most dynamic: as some samples gain weight, they raise $\bar{a}$, making it increasingly difficult for other samples to remain competitive. This may cause a few $w_i$ to dominate $S$ and push other $w_i$ to zero, resulting in a spiky distribution $p$.

We formalize the above intuition by analyzing the behavior of $w_i$ using a "damped" ODE,

$$\dot{w}_i(t) := \frac{1}{S(t)}\left(a_i - \bar{a}(t)\right), \tag{15}$$

$$\frac{dw_i}{dt}(t) := \begin{cases} \dot{w}_i(t), & \text{if } w_i(t) > \delta \text{ or } \dot{w}_i(t) > 0, \\ 0, & \text{if } w_i(t) = \delta \text{ and } \dot{w}_i(t) < 0. \end{cases} \tag{16}$$

The damping of (16) is necessary to prevent negative $w_i$. By modifying the raw gradient $\dot{w}_i(t)$, we ensure that $w_i(t)$ never goes below a floor $\delta \geq 0$ but can leave the floor if $\dot{w}_i(t) > 0$. $\delta$ is a hyperparameter that can be controlled by the activation function of $s_\phi(\cdot)$. Assuming that $(a_i)$ stay constant, and there is a *dominant index $K$* so that $a_K > a_i, \forall i \neq K$, we can show $p_K$ converges to 1, with a formal proof in Appendix B.

**Theorem 3.2** (Convergence of $p$ to a one-hot vector).

$$\lim_{t\to\infty} p_K = 1, \text{ and } \forall i \neq K, \lim_{t\to\infty} p_i = 0. \tag{17}$$

*If $\delta = 0$, $p_K$ reaches 1 in finite time. If $\delta > 0$, the time for $p_K$ to reach $1 - \epsilon$ scales with $O(\epsilon^{-3})$.*

The theorem suggests that, given sufficient training, the network will degenerate to a state where one data point dominates and every other data point contributes very little, if at all, to the training loss. The assumption that $(a_i)$ are constant may seem an oversimplification. However, empirical training dynamics (Figures 2 and 4) are qualitatively consistent with Theorem 3.2, as discussed below.

In reality, $(p_i)$ are not so extreme for several reasons. First, the $a_i$ values are not constant. As the main network $f_\theta(\cdot)$ begins to fit the $i$-th data point, the gradient norm $\|g_i\|_2$ will decrease, causing $a_i$ to decrease and preventing $p_i$ from reaching 1. Second, we usually do not train $s_\phi(\cdot)$ to convergence before switching back to training the main network, which may cause $a_i$ to change. Finally and importantly, when we use a parameterized network $s_\phi(\cdot)$ to predict $w_i$, the network outputs are usually more smooth than a one-hot vector (Ma & Ying, 2021). As a result, $p_K$ usually does not reach 1.

Nevertheless, empirical training dynamics (Figures 2 and 4) are qualitatively consistent with Theorem 3.2. First, even

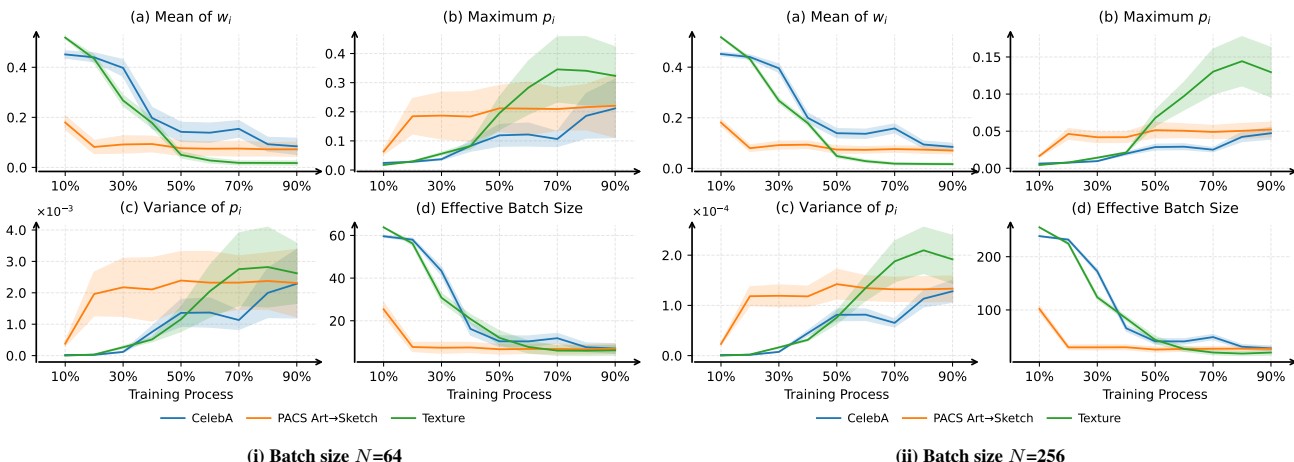

*Figure 2.* Training dynamics across two batch sizes: (a) the mean *unnormalized* weight $\hat{\mathbb{E}}[w_i] = S/N$, which decreases over time, (b) the maximum *normalized* weight $\max_i p_i$, which increases but to a lower ceiling for $N = 256$ (c) the variance $\mathrm{Var}(p_i)$, which increases to a lower ceiling for $N = 256$, and (d) the effective batch size $B_{\mathrm{eff}} = (\sum_i p_i^2)^{-1}$, which is larger for $N = 256$. Colors indicate datasets. Shaded regions show mean $\pm$ std across independent batches. Additional results for $N = 512$ and $N = 1024$ can be found in Figure 4 of Appendix E.

though the distribution of $p$ does not become one-hot, it does become spiky. In Figure 2, one component out of 64 or 256 captures $1/20$ to $1/4$ of the total probability mass. Second, the empirical mean of $w_i$ decreases during training. This is also consistent with the theory, given a relatively smooth $s_\phi(\cdot)$. That is, if we decrease $w_i$ for most $i$, we end up pushing all $w_i$ closer to zero. Third, again as the theory predicts, the sample variance of $p_i$, calculated as $\hat{\sigma}_p^2 := \frac{1}{N}\sum_{i=1}^{N}(p_i - 1/N)^2$, increases. Simultaneously, the effective batch size (EBS), calculated as $\left(\sum_i p_i^2\right)^{-1}$ decreases. This is expected, since

$$\mathrm{EBS} := \left(\sum_i p_i^2\right)^{-1} = \left(\hat{\sigma}_p^2 + \frac{1}{N}\right)^{-1}. \quad (18)$$

Overall, Theorem 3.2 provides important insights into the root cause of the experimental observations. Notably, one may naturally expect that lowering EBS would increase the variance of $\Delta\phi$. We analyze that in the next section.

### 3.4. What Degrades and Repairs GSNR

Theorem 3.2 treats $w_i$ as independent hyperparameters. In this section, we study the complete setting where $(w_i)$ are output of the selection network $s_\phi(\cdot)$ and focus on the selection network gradient $\Delta\phi$ of (10). Our main argument is that a spiky alignment between $\bar{\gamma}$ and $g_i$, as well as a small $S$, collectively lead to the degradation of the GSNR for the selection network. However, increasing batch size $N$ can alleviate this issue.

The GSNR can be seen as a measurement of the quality of

gradient

$$\mathrm{GSNR} := \frac{\left\| \mathbb{E}\left[\Delta\phi\right] \right\|_2^2}{\mathbb{E}\left[\left\|\Delta\phi - \mathbb{E}\left[\Delta\phi\right]\right\|_2^2\right]} = \frac{\left\| \mathbb{E}\left[\Delta\phi\right] \right\|_2^2}{\mathrm{Var}\left(\Delta\phi\right)}, \quad (19)$$

where $\mathrm{Var}(\cdot)$ denotes the total variance, computed as the trace of the covariance matrix. Intuitively, the GSNR measures how closely each stochastic gradient update tracks the direct path toward the local minimum. Thus, an extremely low GSNR can cause convergence difficulties. Further, Liu et al. (2020) show that high GSNR leads to improved generalization.

We capture the spikiness of $p$ with Assumption 3.3. With abuse of notation, we treat $g_i$, $w_i$ and $h_i$ as random variables, so that $a_i = \bar{\gamma}^\top g_i$ is also a random variable. We impose the following condition on the expectation of $(a_i)$. When the expectations are spiky, it is likely for $p_K$ to grow and GSNR to degrade.

**Assumption 3.3** (The expectation of vector $a$ is spiky). There is a unique dominant index $K$ such that $\mu_K^{(a)} > \mu_i^{(a)}$ for all $i \neq K$, where $\mathbb{E}[a_i|w_i, h_i] = \mu_i^{(a)}$.

**Theorem 3.4.** *With Assumption 3.3 and some regularity conditions, $w_K$ has large influence on $\Delta\phi$ with large probability. More specifically, defining the average gap energy $G_{gap} := \sqrt{\frac{1}{N}\sum_{i\neq K}(\mu_K - \mu_i)^2}$,*

$$P\left(\left\|\frac{\partial\Delta\phi}{\partial w_K}\right\|_2 \geq \frac{\sqrt{\alpha}G_{gap}}{S^2}\right) \geq \frac{(\tau_h - \alpha)^2 G_{gap}^4}{K_1 N^9 G_{gap}^4 + K_2 N^8}, \quad (20)$$

*where $\alpha$ is an arbitrary value in $(0, \tau_h)$; $\tau_h$, $K_1$, and $K_2$ are positive constants that do not depend on $N$ or $G_{gap}$.*

This theorem demonstrates that the sensitivity of the gradient $\Delta\phi$ to $w_K$ increases linearly with $G_{\text{gap}}$ and $1/S^2$. Further, the probability of this event approaches 1 as $G_{\text{gap}}$ approaches $\infty$. However, increasing the batch size $N$ will decrease the probability of the event and the lower bound through $S$, since $\mathbb{E}[S] = N\,\mathbb{E}[w_i]$. It is worth noting that increasing $N$ does not increase $G_{\text{gap}}$, which is the average over the training batch.

One natural thought is that, since $\Delta\phi$ is highly sensitive to $w_K$, randomness in $w_K$ may cause high variance in $\Delta\phi$ and low GSNR. To show this, we need to relate any finite change in $w_K$ to the change in $\Delta\phi$. Such a change is created by resampling $(x_K, y_K)$ while keeping all other data points unchanged. This yields a new tuple $(\widetilde{g}_K, \widetilde{w}_K, \widetilde{h}_K)$ and a new gradient of interest $\widetilde{\widetilde{\Delta\phi}}$. With positive probability, the resampling induces a small change to $w_K$, by a bounded second-order derivative and a Taylor-expansion argument, we can show that the ratio

$$L_K := \frac{\|\widetilde{\widetilde{\Delta\phi}} - \Delta\phi\|_2}{|\widetilde{w}_K - w_K|}, \tag{21}$$

on the event $\widetilde{w}_K \neq w_K$, is bounded from below, as a direct corollary to Theorem 3.4.

**Corollary 3.5** (Local Lipschitz lower bound under resampling, informal)**.** *There exists a constant $K_3 > 0$ depending only on the distribution of $(g_i, w_i, h_i)$ and the validation gradient $\bar{\gamma}$, but not on $N$ or $G_{gap}$,*

$$P\left(L_K \geq K_3\,\frac{\sqrt{\alpha}\,G_{gap}}{S^2}\right) > 0, \tag{22}$$

*where $\alpha$ is the same constant in Theorem 3.4.*

The main result follows from the lower bound on $L_K$. In plain words, at any snapshot of $\phi$ and $\theta$, due to the variance induced by random sampling of the data point at the dominant index, $(x_K, y_K)$, the GSNR becomes low. We establish an upper bound on GSNR that involves $N$.

**Theorem 3.6** (GSNR upper bound, informal)**.**

$$\text{GSNR} \leq C_{\text{SNR}}\,\frac{N^4\,\mathbb{E}[S^{-2}]}{G_{gap}^2\,\mathbb{E}\left[S^{-4}\mathbb{1}_{E_{Lip}}\right]}, \tag{23}$$

*where constant $C_{\text{SNR}} > 0$ is independent of $N$ and $G_{gap}^2$. $\mathbb{1}_{E_{Lip}}$ equals 1 if the lower bound on $L_K$ from Corollary 3.5 holds and 0 otherwise.*

The theorem indicates that a small $S$ and a large $G_{\text{gap}}$ degrade GSNR. A small $S$ is the implication of Theorem 3.2 and their practical existence is confirmed by Figure 2.

The theory also hints at an obvious solution: increasing the batch size $N$. Doing so has the effect of increasing $N^4$ and increasing $S$, since $\mathbb{E}[S] = N\,\mathbb{E}[w_i]$. This solution

aligns well with empirical results of Figures 1, 2, and 4. In particular, according to Theorem 3.2, spiky $a_i$ should cause degeneration in data selection and a large $\max_i p_i$. However, Figures 2 and 4 show that this phenomenon is alleviated by a large batch size.

## 4. Input Features for the Selection Network

An effective GSNR ensures that we can learn the selection network. However, the network can only be effective at selecting data if it has good features that contain information about data quality. To this end, we devise a rich feature vector that encodes a wide range of signals related to generalization, including local, global, optimization dynamics, and class-indicator features. We assume the availability of a small validation set, whose distribution is similar to the true test, during training. Due to space constraints, we discuss the features briefly here and leave details to Appendix I.

*Local features* characterize the local geometry around each input $x_i$ through its cosine similarities to $K$ nearest neighbors (KNN) in the training and validation sets. Intuitively, if $x_i$ is very similar to many training data points, it may be redundant. Conversely, if it is far away from most training or validation instances, it may be an outlier that does not contribute to generalization.

Formally, let $\mathcal{N}_K(x_i; \mathcal{A})$ denote the $K$ nearest neighbors of $x_i$ in the set $\mathcal{A}$. We use the features $\{\cos(h_i, h_j) \mid j \in \mathcal{N}_K(x_i; \mathcal{A})\}$, where $\mathcal{A}$ can be the current training batch, the validation batch, the training set, and the validation set. We use online encodings $h^{\text{cls}}$ from the current classifier $f_\theta(\cdot)$ for batch-level KNN, and offline encodings $h^{\text{gen}}$ from an image generation model for dataset-level KNN. Lastly, to identify potential semantic inconsistencies, we compute the label agreement (Zhang et al., 2024) among the $K$ nearest neighbors in $\mathcal{D}_{\text{train}}$ using $h^{\text{gen}}$, defined as $\frac{1}{K}\sum_{j \in \mathcal{N}_K(x_i; \mathcal{D}_{\text{train}})} \mathbb{1}[y_j = y_i]$, where $\mathbb{1}[\cdot]$ is the indicator function.

*Global features* are computed as the Euclidean distance and cosine similarity between the encoding of $x_i$ and its class centers in the training and the validation sets, respectively. These features reveal the typicality of $x_i$ within the class $y_i$. Intuitively, data points at the fringe of the source distribution, or that are prototypical in the target distribution, can be important for transfer learning.

Formally, let $\mathcal{A}_c$ denote the subset of samples with class $c$ in set $\mathcal{A}$. Using offline encodings $h^{\text{gen}}$ from an image generation model, we compute the class centroid as $\mu_{\mathcal{A}_c} = \frac{1}{|\mathcal{A}_c|}\sum_{x_j \in \mathcal{A}_c} h_j^{\text{gen}}$. For each sample $x_i$ with label $y_i$, we use its $\ell_2$ distance and cosine similarity to the corresponding class centroid, $\{\|h_i^{\text{gen}} - \mu_{\mathcal{A}^{y_i}}\|_2,\ \cos(h_i^{\text{gen}}, \mu_{\mathcal{A}^{y_i}})\}$, as global features, where $\mathcal{A}$ can be either the training set or the validation set.

*Optimization dynamics features* characterize how the data point interacts with the optimization process. We train a separate neural classifier without data selection to extract the following features: (i) the number of forgetting events (Toneva et al., 2019) to track longitudinal training stability, and (ii) the gradient norm and error vector norm (Paul et al., 2021) to measure sample difficulty.

*Class-indicator features* provide explicit categorical context. We include learnable embeddings for the class label and for the data source (real vs. synthetic), enabling the selection network to distinguish class-specific and generator-related patterns.

## 5. Experiments

**Datasets.** We evaluate on four benchmarks: Waterbirds (Sagawa et al., 2020), CelebA (Yuan et al., 2024), Texture (Geirhos et al., 2019), and PACS (Li et al., 2017). PACS contains 4 domains: Art Painting, Cartoon, Photo, and Sketch. We perform pairwise transfer learning with a total of 12 pairs of domains. For each benchmark, the training split differs from the validation and test splits, which share the same target distribution. Following Yuan et al. (2024) and Dunlap et al. (2023), we augment the training data with roughly equal number of synthetic images. Details are in Appendix F.

**Baselines.** We compare against a diverse set of data selection methods spanning four categories: (i) training-dynamics heuristics, including *Forgetting Events* (Toneva et al., 2019) and *GraNd* (Paul et al., 2021). (ii) synthetic image selection heuristics, including *CLIP Similarity* (Zhao et al., 2024; Li et al., 2024), *Auxiliary Classifier* (Aux-Clf) (Lampis et al., 2023), and a stronger variant of Aux-Clf, *Aux-Clf (Val)*, which we trained on the validation set to provide a more reliable target-distribution alignment signal, (iii) domain adaptation methods, including *NormSim* (Wang et al., 2024a) and *Importance Weighting* (Choi et al., 2021), (iv) a meta-learning method, *MetaWeightNet* (Shu et al., 2019). For every method except *Importance Weighting* and *MetaWeightNet*, we create a soft selection version, which uses the data score as weight, and a hard selection version, which thresholds the score into 0/1 weights. Details of these baselines are in Appendix G.

**Implementation.** The classification network is ResNet-50, which is initialized with ImageNet-pretrained weights and optimized by SGD with momentum. The selection network is a three-layer MLP with hidden dimension 100 and sigmoid activation function, optimized with AdamW. We use a batch size of $N = 1024$, and set $K = 3$ for $K$ nearest neighbors in input feature computation. Data generation and classifier training follow the original settings in Dunlap et al. (2023); Yuan et al. (2024). Further details

are shown in Appendix H.

### 5.1. Main Results

Table 1 reports the performance of all methods across four benchmarks. We summarize the key observations below.

**Our method achieves consistently large gains.** Our method attains the highest average accuracy of 73.75%, improving upon No Selection by 5.49% and the strongest baseline, hard selection with Aux-Clf (Val), by 2.89%.

**MetaWeightNet performs inconsistently.** MetaWeightNet is the most similar to our work, but it differs by (1) using a batch size of 64, (2) using online loss as the only input feature, and (3) does not normalize the output weights. It achieves good performance on Texture (+3.37%) and PACS-Photo (+4.08%) but not other domains. Overall, it only achieves a modest 0.62% average improvement over No Selection. These results corroborates our argument that simplistic application of MTS often underperforms.

**Training dynamics methods and CLIP similarity do not improve performance.** Training-dynamics heuristics, including Forgetting Events and GraNd, underperform the No Selection baseline, with average accuracy drops of 0.53%-1.37%. Similarly, CLIP Similarity yields only marginal changes between from $-0.17\%$ to $+0.02\%$ relative to No Selection.

**Auxiliary classifier–based selection exhibits sharply different behaviors depending on its training domain.** When trained on training data, Aux-Clf results in substantial performance drops of 3.63%-5.39%. However, Aux-Clf (Val), trained on validation data, achieves improvements of up to 2.60% under Hard Selection and is the best baseline. These results suggest that Aux-Clf is highly dependent on the domain it is trained on and may amplifying domain disparity.

**Domain adaptation methods exhibit dataset-specific behavior.** For instance, *NormSim* improves performance on photorealistic datasets such as WaterBirds and CelebA, but degrades accuracy by up to 8.96% on PACS-Sketch. This discrepancy stems from its reliance on pretrained CLIP embeddings (Radford et al., 2021), which are well-suited to natural images but less reliable for visually atypical domains such as sketches. Similarly, *Importance Weight* yields uneven results, with a mean performance drop of 0.91%. This instability aligns with known challenges in density-ratio estimation, where limited overlap between source and target distributions can lead to high-variance estimates and unreliable reweighting.

### 5.2. Ablation Experiments

We conduct ablation experiments to analyze the impact of batch size and input feature design. For input features, we

*Table 1.* Main results. Hard selection discards samples before training, while soft selection reweights samples during training.

| Method | WaterBirds | CelebA | Texture | PACS | | | | Average |
|---|---|---|---|---|---|---|---|---|
| | | | | Art | Cartoon | Photo | Sketch | |
| No Selection | 70.13 | 78.52 | 24.44 | 82.76 | 85.23 | 60.08 | 76.62 | 68.26 |
| *Hard Selection Methods* | | | | | | | | |
| Forgetting (Toneva et al., 2019) | 70.07 | 78.45 | 25.03 | 81.42 | 83.63 | 56.63 | 72.99 | 66.89 -1.37 |
| GraNd (Paul et al., 2021) | 67.82 | 81.01 | 24.06 | 81.79 | 84.93 | 59.52 | 73.90 | 67.58 -0.68 |
| CLIP Similarity (Li et al., 2024) | 69.96 | 78.00 | 25.17 | 83.00 | 84.67 | 58.87 | 78.28 | 68.28 +0.02 |
| Aux-Clf (Dunlap et al., 2023) | 67.01 | 77.37 | 21.81 | 77.59 | 82.74 | 46.97 | 66.57 | 62.87 -5.39 |
| Aux-Clf (Val) (Dunlap et al., 2023) | 78.07 | 84.30 | 27.47 | 82.13 | 85.54 | 61.58 | 76.96 | 70.86 +2.60 |
| NormSim (Wang et al., 2024a) | 70.84 | 78.63 | 24.76 | 82.83 | 84.13 | 60.24 | 74.99 | 68.06 -0.20 |
| *Soft Selection Methods* | | | | | | | | |
| Forgetting (Toneva et al., 2019) | 69.88 | 78.63 | 24.27 | 81.68 | 85.39 | 58.23 | 73.72 | 67.40 -0.86 |
| GraNd (Paul et al., 2021) | 69.44 | 80.24 | 23.51 | 81.89 | 85.34 | 59.10 | 74.57 | 67.73 -0.53 |
| CLIP Similarity (Li et al., 2024) | 70.19 | 78.42 | 24.37 | 82.37 | 85.24 | 59.95 | 76.12 | 68.09 -0.17 |
| Aux-Clf (Dunlap et al., 2023) | 64.04 | 76.98 | 21.60 | 80.16 | 84.74 | 54.28 | 70.6 | 64.63 -3.63 |
| Aux-Clf (Val) (Dunlap et al., 2023) | 78.91 | 84.47 | 26.67 | 82.56 | 85.88 | 63.20 | 69.13 | 70.12 +1.86 |
| NormSim (Wang et al., 2024a) | 70.22 | 79.05 | 24.48 | 82.36 | 85.05 | 61.09 | 67.66 | 67.13 -1.13 |
| Importance Weight (Choi et al., 2021) | 70.13 | 75.97 | 25.73 | 80.86 | 84.70 | 60.64 | 73.44 | 67.35 -0.91 |
| MetaWeightNet (Shu et al., 2019) | 72.24 | 76.04 | 27.81 | **83.56** | 85.41 | 64.16 | 72.68 | 68.84 +0.62 |
| **Ours** | **81.81** | **85.01** | **29.38** | 83.07 | **86.79** | **71.16** | **79.06** | **73.75** +5.49 |

*Table 2.* Ablation results showing the impact of batch size and input feature design on selection performance.

| Method | WaterBirds | CelebA | Texture | PACS | | | | Average |
|---|---|---|---|---|---|---|---|---|
| | | | | Art | Cartoon | Photo | Sketch | |
| No Selection | 70.13 | 78.52 | 24.44 | 82.76 | 85.23 | 60.08 | 76.62 | 68.26 |
| Ours ($N = 1024$) | 81.81 | **85.01** | 29.38 | **83.07** | **86.79** | **71.16** | **79.06** | **73.75** |
| *Decreasing Batch Size N to ...* | | | | | | | | |
| $N = 256$ | 81.64 | 84.68 | 27.81 | 82.86 | **86.79** | 67.32 | 77.66 | 72.68 -1.07 |
| $N = 64$ | **81.82** | 80.27 | 28.33 | 82.27 | 86.02 | 64.40 | 73.62 | 70.96 -2.79 |
| *Changing Input Features to ...* | | | | | | | | |
| Raw Image | 78.78 | 73.63 | 24.31 | 80.53 | 85.30 | 60.44 | 74.38 | 68.20 -5.55 |
| ResNet Features | 80.60 | 83.39 | **29.72** | 82.23 | 86.19 | 67.82 | 72.47 | 71.77 -1.95 |
| Training Loss | 72.63 | 78.28 | 28.96 | 82.44 | 85.09 | 63.45 | 78.10 | 69.85 -3.90 |

consider three alternatives: (1) *Raw Image*, where the selection network operates directly on pixel values; (2) *ResNet Features*, which use the online classifier's penultimate-layer embeddings; and (3) *Training Loss*, which relies solely on the scalar per-sample loss. We draw the following observations. The results are shown in Table 2.

**Selection performance strongly depends on batch size.** Compared to our default setting ($N = 1024$), reducing the batch size to 256 leads to an average performance degradation of 1.07%, which further increases to 2.79% when $N = 64$. This degradation is particularly pronounced on PACS (Photo and Sketch) and CelebA, where accuracy drops by up to 6.76%. The strong empirical correlation between performance and batch size supports our theoretical analysis, demonstrating that larger batch sizes are crucial for effective MTS.

**The proposed feature set outperforms ResNet features and the training loss feature.** Both ResNet Features and

Training Loss lag behind the proposed feature design by 1.95%-3.90%. These results demonstrate the necessity to utilize informative features that correlate well with data quality. As our features include distributional information involving the relations between data points, they may not be easily inferred from the feature of a single data point.

**Data selection works poorly on raw inputs.** The raw image features result in the largest performance degradation, with an average drop of 5.55%. This is partially attributed to the shallow architecture of the selection network, which could not automatically learn high-quality representations from images directly. However, increasing the parameter count of the selection network would result in substantially increased computational overhead, since computing $\Delta\phi$ requires two computing and storing two gradient vectors of the same size as $\phi$. As a result, scaling to a much larger selection network can be difficult (Liu et al., 2018).

Overall, these ablations demonstrate that both large batch

*Table 3.* Performance under the few-shot validation setting. In the 5-shot setting, each class in the validation set contains only 5 samples. Full-shot results are shown in italics for reference.

| Setting | Method | Waterbirds | CelebA | Texture | Avg. |
|---------|--------|-----------|--------|---------|------|
| | No Selection | 70.31 | 78.52 | 24.44 | 57.76 |
| 5-shot | MW-Net | 67.55 | 78.59 | 24.89 | 57.01 |
| 5-shot | Aux-Clf (Val) | 72.78 | 78.77 | 24.20 | 58.58 |
| 5-shot | Ours | **77.51** | **81.53** | **25.41** | **61.48** |
| *Full-shot* | *Ours* | *81.81* | *85.01* | *29.38* | *65.40* |

*Table 4.* Results with a ViT-Base-Patch32 backbone pretrained on ImageNet-21K.

| Method | Waterbirds | CelebA | Texture | Avg. |
|--------|-----------|--------|---------|------|
| No Selection | 76.93 | 82.79 | 39.17 | 66.30 |
| MW-Net | 76.72 | 80.90 | 39.90 | 65.84 |
| Aux-Clf (Val) | 79.22 | **86.36** | 44.34 | 69.97 |
| Ours | **82.19** | 85.21 | **47.40** | **71.60** |

sizes and informative feature representations are critical for stable and effective selection-network training.

### 5.3. Few-shot Validation Data

We evaluate our method in a few-shot validation-set setting, where each class in the validation set contains only 5 samples. As shown in Table 3, the average performance drops by 3.92% compared with the full-shot validation setting, highlighting the importance of validation data quantity for MTS. Nevertheless, even under this highly constrained setting, our method still outperforms No Selection by 3.72% and the strongest baseline, Aux-Clf (Val), by 2.90% on average.

### 5.4. Generalization across Architectures

To examine whether our method generalizes beyond a specific backbone architecture, we conduct experiments using ViT-Base-Patch32 pretrained on ImageNet-21K [2] as the visual backbone. As shown in Table 4, our method achieves the best average performance among all methods. In particular, it improves over No Selection and MW-Net by 5.30% and 5.76% on average, respectively. These results suggest that our method is architecture-agnostic.

### 5.5. Accuracy-memory Trade-off

One limitation of a large batch is the the high memory footprint incurred. In Figure 3, we visualize the computational trade-off between accuracy and memory usage, showing the average accuracy values from Tables 1 and 2. The memory usage is shown as the area of the circles. As the batch size $N$ increases, classification accuracy improves at the

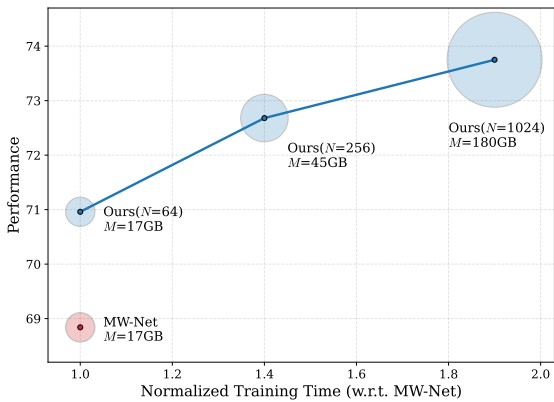

*Figure 3.* Accuracy-cost trade-off across different batch sizes $N$. The memory usage is visualized as the area of the circles. The accuracy values reported are the average accuracy from 1 and 2.

cost of higher memory usage. At a moderate batch size of $N = 256$, which fits into a single NVIDIA RTX A6000 GPU, our method improves over MW-Net by 3.84%.

We note there are computational techniques that increase batch sizes without increasing memory usage. For example, gradient accumulation[3] takes the average gradients from several backward passes for each parameter update. Other techniques such as mixed-precision training and gradient checkpointing may also reduce memory usage. We leave the exploration of those techniques as future work.

## 6. Conclusion

In this work, we analyze Meta-learning Training-data Selection (MTS) and identify two key obstacles that limit its practical effectiveness, including a low gradient signal-to-noise ratio (GSNR) and lack of informative features correlated to data quality. Theoretical analysis reveals the deep cause of the low GSNR and reveals that increasing the batch size provides a simple and effective remedy. In addition, we propose a set of informative distributional and training-dynamics features that correlate well with data usefulness. By jointly addressing these issues, our approach significantly improves the performance of MTS on synthetic data selection, achieving consistent gains across four datasets and fifteen settings, and outperforming both training without data selection and strong heuristic baselines.

## Impact Statement

This paper presents work whose goal is to advance the field of machine learning. There are many potential societal consequences of our work, none of which we feel must be

---

[2] https://huggingface.co/google/vit-base-patch32-224-in21k

[3] https://huggingface.co/docs/transformers/grad_accumulation

specifically highlighted here.

## Acknowledgment

This research is supported, in part, by the RIE2025 Industry Alignment Fund – Industry Collaboration Projects (IAF-ICP) (Award I2301E0026), administered by A*STAR, as well as supported by Alibaba Group and NTU Singapore through Alibaba-NTU Global e-Sustainability CorpLab (ANGEL). The research is also partially funded by National Research Foundation Fellowship (NRFF13-2021-0006), Singapore.

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

## Appendix Outline

# A. Derivation of the Hypergradient Expressions

## A.1. Proof of Lemma 3.1

**Lemma 3.1.** *The update to $\phi$ is computed as*

$$\phi_{t+1} = \phi_t - \eta_\theta \eta_\phi \Delta \phi_t, \tag{9}$$

*where the hypergradient $\Delta \phi_t$ is*

$$\Delta \phi := -\frac{1}{S} \left( \frac{1}{N} \sum_{i=1}^{N} \gamma_i(\theta'_t) \right)^\top \sum_{i=1}^{N} g_i(\theta_t) \left( h_i(\phi_t)^\top - \frac{w_i}{S} \sum_{j=1}^{N} h_j(\phi_t)^\top \right), \tag{10}$$

*$\theta'_t$ is the result of the one-step update to $\theta$ inside (7),*

$$\theta'_t = \theta_t - \frac{\eta_\theta}{S} \sum_{i=1}^{N} w_i g_i(\theta_t), \tag{11}$$

*and $\eta_\theta$ and $\eta_\phi$ are learning rates.*

*Proof.* We make explicit the dependence of $w_i$ and $S$ on $\phi$:

$$w_i(\phi) := w(x_i, \phi), \quad \ell_i(\theta) := \ell\big(f(x_i, \theta), y_i\big), \quad S(\phi) := \sum_{i=1}^{N} w_i(\phi). \tag{24}$$

The training and validation losses are

$$\mathcal{L}_{\text{train}}(\theta, \phi) := \frac{1}{S(\phi)} \sum_{i=1}^{N} w_i(\phi)\, \ell_i(\theta), \tag{25}$$

$$\mathcal{L}_{\text{val}}(\theta) := \frac{1}{N} \sum_{i=1}^{N} \ell\big(f(x_i^{\text{v}}, \theta), y_i^{\text{v}}\big). \tag{26}$$

Our goal is to compute the total derivative $\mathrm{d}\mathcal{L}_{\text{val}}(\theta'_t)/\mathrm{d}\phi_t$, where $\theta'_t$ is obtained by a single gradient step on the training loss:

$$\theta'_t = \theta_t - \eta_\theta\, \nabla_\theta \mathcal{L}_{\text{train}}(\theta_t, \phi_t), \tag{27}$$

with learning rate $\eta_\theta > 0$.

For notational convenience, define

$$g_i(\theta_t) := \nabla_\theta \ell_i(\theta_t) = \nabla_\theta \ell\big(f(x_i, \theta_t), y_i\big), \tag{28}$$

$$h_i(\phi_t) := \nabla_\phi w_i(\phi_t) = \nabla_\phi w(x_i, \phi_t), \tag{29}$$

$$g_i^{\text{v}}(\theta_t) := \nabla_\theta \ell\big(f(x_i^{\text{v}}, \theta_t), y_i^{\text{v}}\big), \tag{30}$$

$$g^{\text{v}}(\theta_t) := \nabla_\theta \mathcal{L}_{\text{val}}(\theta_t) = \frac{1}{N} \sum_{i=1}^{N} g_i^{\text{v}}(\theta_t). \tag{31}$$

We treat gradients as column vectors.

In (25), only $\ell_i(\theta)$ depends on $\theta$, hence

$$\nabla_\theta \mathcal{L}_{\text{train}}(\theta_t, \phi_t) = \frac{1}{S(\phi_t)} \sum_{i=1}^{N} w_i(\phi_t)\, g_i(\theta_t). \tag{32}$$

Differentiating (32) w.r.t. $\phi_t$, we obtain

$$\frac{\partial}{\partial \phi_t}\big(\nabla_\theta \mathcal{L}_{\text{train}}(\theta_t, \phi_t)\big) = \frac{1}{S(\phi_t)}\sum_{i=1}^{N} g_i(\theta_t)\, h_i(\phi_t)^\top - \frac{1}{S(\phi_t)^2}\Big(\sum_{i=1}^{N} w_i(\phi_t)\, g_i(\theta_t)\Big)\Big(\sum_{j=1}^{N} h_j(\phi_t)\Big)^\top$$

$$= \frac{1}{S(\phi_t)}\sum_{i=1}^{N} g_i(\theta_t)\Big(h_i(\phi_t) - \frac{w_i(\phi_t)}{S(\phi_t)}\sum_{j=1}^{N} h_j(\phi_t)\Big)^\top. \tag{33}$$

The validation loss after the inner update is

$$\mathcal{L}_{\text{val}}(\theta_t') = \mathcal{L}_{\text{val}}\big(\theta_t'(\phi_t)\big), \qquad \theta_t'(\phi_t) = \theta_t - \eta_\theta\, \nabla_{\theta_t}\mathcal{L}_{\text{train}}(\theta_t, \phi_t). \tag{34}$$

By the chain rule,

$$\frac{d}{d\phi_t}\mathcal{L}_{\text{val}}(\theta_t') = \big(g^{\text{v}}(\theta_t')\big)^\top \frac{d\theta_t'}{d\phi_t}, \tag{35}$$

and since $\theta_t'(\phi_t) = \theta_t - \eta_\theta \nabla_\theta \mathcal{L}_{\text{train}}(\theta_t, \phi_t)$,

$$\frac{d\theta_t'}{d\phi_t} = -\eta_\theta\, \frac{\partial}{\partial \phi_t}\big(\nabla_\theta \mathcal{L}_{\text{train}}(\theta_t, \phi_t)\big). \tag{36}$$

Substituting into (35) and using (33) gives

$$\frac{d}{d\phi_t}\mathcal{L}_{\text{val}}(\theta_t') = -\eta_\theta\big(g^{\text{v}}(\theta_t')\big)^\top \left[\frac{1}{S(\phi_t)}\sum_{i=1}^{N} g_i(\theta_t)\Big(h_i(\phi_t) - \frac{w_i(\phi_t)}{S(\phi_t)}\sum_{j=1}^{N} h_j(\phi_t)\Big)^\top\right]. \tag{37}$$

Finally, writing $g^{\text{v}}(\theta_t') = \frac{1}{N}\sum_{i=1}^{N} g_i^{\text{v}}(\theta_t')$ yields the stated expression for $\Delta\phi$. $\qquad\square$

### A.2. Hypergradient of $\mathcal{L}_{\text{val}}$ w.r.t. $w_i$

We consider the dependence of the post-update validation loss on the individual training weight $w_i$. Recall that the training objective is

$$\mathcal{L}_{\text{train}}(\theta, w) = \frac{1}{S}\sum_k w_k\, \ell_k(\theta), \qquad S = \sum_k w_k, \tag{38}$$

where $\ell_k(\theta) = \ell(f(x_k, \theta), y_k)$ and $w_k = w(x_k, \phi)$. Let $g_k := \nabla_\theta \ell_k(\theta)$ denote the per-example training gradients. A single gradient step on $\theta$ with learning rate $\eta_\theta$ yields

$$\theta' = \theta - \eta_\theta \nabla_\theta \mathcal{L}_{\text{train}}(\theta, w) = \theta - \eta_\theta \frac{1}{S}\sum_k w_k g_k. \tag{39}$$

We evaluate the validation loss at $\theta'$,

$$\mathcal{L}_{\text{val}}(\theta') = \frac{1}{N}\sum_{j=1}^{N} \ell(f(x_j, \theta'), y_j), \tag{40}$$

and denote its gradient by

$$\gamma' := \nabla_\theta \mathcal{L}_{\text{val}}(\theta'). \tag{41}$$

Using the chain rule, the total derivative of $\mathcal{L}_{\text{val}}$ with respect to an individual weight $w_i$ is

$$\frac{d\mathcal{L}_{\text{val}}(\theta')}{dw_i} = \Big(\frac{\partial \mathcal{L}_{\text{val}}}{\partial \theta'}\Big)^\top \frac{\partial \theta'}{\partial w_i} = (\gamma')^\top \frac{\partial \theta'}{\partial w_i}. \tag{42}$$

From the update rule for $\theta'$,

$$\theta' = \theta - \eta_\theta \frac{1}{S}\sum_k w_k g_k \quad \Rightarrow \quad \frac{\partial \theta'}{\partial w_i} = -\eta_\theta \frac{\partial}{\partial w_i}\Big(\frac{1}{S}\sum_k w_k g_k\Big). \tag{43}$$

$$\frac{\partial}{\partial w_i}\left(\frac{1}{S}\sum_k w_k g_k\right) = \frac{1}{S}\frac{\partial \sum_k w_k g_k}{\partial w_i} - \frac{\sum_k w_k g_k}{S^2}\frac{\partial S}{\partial w_i} = \frac{Sg_i - \sum_k w_k g_k}{S^2}, \tag{44}$$

as $\frac{\partial}{\partial w_i}\sum_k w_k g_k = g_i$ and $\frac{\partial S}{\partial w_i} = 1$. Hence,

$$\frac{\partial \theta'}{\partial w_i} = -\eta_\theta \frac{Sg_i - \sum_k w_k g_k}{S^2}, \tag{45}$$

$$\frac{d\mathcal{L}_{\text{val}}}{dw_i} = (\gamma')^\top\left(-\eta_\theta \frac{Sg_i - \sum_k w_k g_k}{S^2}\right) = -\eta_\theta \frac{1}{S}(\gamma')^\top\left[g_i - \frac{1}{S}\sum_k w_k g_k\right]. \tag{46}$$

Since $p_i = w_i/S$,

$$\frac{d\mathcal{L}_{\text{val}}}{dw_i} = -\eta_\theta \frac{1}{S}(\gamma')^\top\left[g_i - \sum_k p_k g_k\right]. \tag{47}$$

## B. Theorem 3.2 on the Dynamics of Normalized Data Weights $p_i$

**Problem Setup.** Let $N \geq 2$ and let $a_1, \ldots, a_N \in \mathbb{R}$ be fixed constants. Assume there is a unique dominant index $K \in \{1, \ldots, N\}$ with

$$a_K > a_i \quad \text{for all } i \neq K. \tag{48}$$

Let $t \geq 0$ denote the continuous time variable. Consider continuous functions $w_i(t) \in [\delta, \infty)$, $i = 1, \ldots, N$, where $\delta \geq 0$. $w_i(t)$ satisfy the following "clamped" dynamics.

Define, for all $t$,

$$S(t) := \sum_{j=1}^N w_j(t) > 0, \qquad p_i(t) := \frac{w_i(t)}{S(t)}, \qquad \bar{a}(t) := \sum_{j=1}^N p_j(t)a_j, \quad \dot{w}_i(t) = \frac{1}{S(t)}\big(a_i - \bar{a}(t)\big) \tag{49}$$

For all $t \geq 0$ and every $i \in \{1, \ldots, N\}$,

$$\frac{dw_i}{dt}(t) = \begin{cases} \dot{w}_i(t), & \text{if } w_i(t) > \delta \text{ or } \dot{w}_i(t) > 0, \\ 0, & \text{if } w_i(t) = \delta \text{ and } \dot{w}_i(t) < 0. \end{cases} \tag{50}$$

Equivalently, $\frac{dw_i}{dt}(t) = \dot{w}_i(t)$ if and only if $i \in A(t)$, where $A(t)$ is the set of active indices:

$$A(t) := \Big\{i : w_i(t) > \delta \text{ or } \big(w_i(t) = \delta \text{ and } a_i > \bar{a}(t)\big)\Big\}. \tag{51}$$

In plain English, each $w_i$ follows the original ODE $\frac{1}{S}(a_i - \bar{a})$ whenever it is not being pushed below the threshold $\delta$. If $w_i(t) = \delta$ and the ODE pushes it higher, it is allowed to leave the floor. Otherwise, $w_i(t)$ is clamped at $w_i(t) = \delta$.

We assume $\forall i, w_i(0) > \delta$ so that $S(0) > 0$.

**Theorem 3.2** (Convergence of normalized weights with lower bound). *With the problem defined above, the following hold.*

*(1) (Convergence of p) For any $\delta \geq 0$, the normalized weights converge to a one-hot vector with 1 at the dominant index:*

$$\lim_{t\to\infty} p_K(t) = 1, \qquad \lim_{t\to\infty} p_i(t) = 0 \quad \text{for all } i \neq K. \tag{52}$$

*(2) (Finite-time convergence when $\delta = 0$) If $\delta = 0$, there exists a finite time $T_* \geq 0$ such that*

$$p_K(t) = 1 \quad \text{and} \quad p_i(t) = 0 \quad (i \neq K) \qquad \text{for all } t \geq T_*. \tag{53}$$

*In other words, the vector $p(t)$ becomes exactly one-hot in finite time.*

*(3) (Polynomial-rate convergence when $\delta > 0$) If $\delta > 0$, then there exist constants $T_0, C_1, C_2 > 0$ (depending on $a_i$, $\delta$, and $w_i(0)$) such that*

$$1 - p_K(t) \leq \frac{C_1}{(t - T_0)^{1/3}} \quad \text{for all } t \geq T_0. \tag{54}$$

*In particular, for every sufficiently small $\epsilon \in (0, 1)$ there exists a time $T_\epsilon \leq T_0 + C_2\, \epsilon^{-3}$ such that*

$$p_K(t) \geq 1 - \epsilon \quad \text{for all } t \geq T_\epsilon. \tag{55}$$

*Proof.* Throughout the proof we write $S(t) := \sum_i w_i(t)$, $p_i(t) := w_i(t)/S(t)$, and $\bar{a}(t) := \sum_j p_j(t)a_j$. We denote $a_{\min} := \min_i a_i$ and $a_{\max} := a_K$. $\bar{a}(t)$ is a convex combination of $a_i$, so $a_{\min} \leq \bar{a}(t) \leq a_{\max}$.

Since $w_K(0) > \delta$ and, whenever $w_K(t) > \delta$, we have

$$\frac{dw_K}{dt}(t) = \frac{a_K - \bar{a}(t)}{S(t)} \geq 0 \quad \text{(because } \bar{a}(t) \leq a_K), \tag{56}$$

it follows that $w_K(t)$ is nondecreasing and

$$w_K(t) \geq w_K(0) > \delta \quad \Rightarrow \quad S(t) \geq w_K(t) > 0 \tag{57}$$

for all $t \geq 0$. Thus $S(t)$ and $\bar{a}(t)$ are well-defined for all $t$.

### Step 1. Monotonicity and convergence of $\bar{a}(t)$.

By definition,

$$\bar{a}(t) = \frac{1}{S(t)} \sum_{i=1}^{N} w_i(t)a_i, \tag{58}$$

so $\bar{a}(t)$ is always a convex combination of the $a_i$ and hence

$$a_{\min} \leq \bar{a}(t) \leq a_{\max} = a_K \quad \text{for all } t \geq 0. \tag{59}$$

Differentiating and using (50), we obtain

$$\frac{d\bar{a}}{dt} = \frac{1}{S} \sum_i a_i \frac{dw_i}{dt} - \frac{\bar{a}}{S} \frac{dS}{dt}. \tag{60}$$

Only indices in $A(t)$ have nonzero derivatives, so

$$\sum_i a_i \frac{dw_i}{dt} = \sum_{i \in A(t)} a_i \frac{1}{S}(a_i - \bar{a}), \qquad \frac{dS}{dt} = \sum_{i \in A(t)} \frac{1}{S}(a_i - \bar{a}). \tag{61}$$

Therefore

$$\frac{d\bar{a}}{dt} = \frac{1}{S^2} \sum_{i \in A(t)} a_i(a_i - \bar{a}) - \frac{\bar{a}}{S^2} \sum_{i \in A(t)} (a_i - \bar{a}). \tag{62}$$

For each $i \in A(t)$, $a_i(a_i - \bar{a}) - \bar{a}(a_i - \bar{a}) = (a_i - \bar{a})^2$. Hence

$$\frac{d\bar{a}}{dt} = \frac{1}{S(t)^2} \sum_{i \in A(t)} (a_i - \bar{a}(t))^2 \geq 0. \tag{63}$$

Thus $\bar{a}(t)$ is nondecreasing and bounded above by $a_K$, so there is a limit

$$\bar{a}_\infty := \lim_{t \to \infty} \bar{a}(t) \in [a_{\min}, a_K]. \tag{64}$$

### Step 2. A linear growth bound on $S^2(t)$ and divergence of $\int \dfrac{dt}{S^2}$ and $\int \dfrac{dt}{S}$.

From (50),

$$\frac{dS}{dt} = \sum_{i \in A(t)} \frac{1}{S}(a_i - \bar{a}), \tag{65}$$

hence

$$\frac{d(S^2)}{dt} = 2S\frac{dS}{dt} = 2\sum_{i\in A(t)}(a_i - \bar{a}(t)). \tag{66}$$

We have $|a_i - \bar{a}(t)| \leq a_{\max} - a_{\min}$ and $|A(t)| \leq N$, so

$$\left|\frac{d(S^2)}{dt}\right| \leq 2N\,(a_{\max} - a_{\min}). \tag{67}$$

Define $C_1 := S^2(0)$ and $C_2 := 2N\,(a_{\max} - a_{\min}) > 0$. By integrating both sides of $\frac{d(S^2)}{dt} \leq 2N\,(a_{\max} - a_{\min})$:

$$S^2(t) \leq S^2(0) + C_2 t = C_1 + C_2 t \quad \text{for all } t \geq 0. \tag{68}$$

Thus

$$S(t) \leq \sqrt{C_1 + C_2 t} \quad \Rightarrow \quad \frac{1}{S(t)^2} \geq \frac{1}{C_1 + C_2 t}, \quad \frac{1}{S(t)} \geq \frac{1}{\sqrt{C_1 + C_2 t}}. \tag{69}$$

Consequently

$$\int_0^\infty \frac{dt}{S(t)^2} \geq \int_0^\infty \frac{dt}{C_1 + C_2 t} = \infty, \tag{70}$$

and similarly

$$\int_0^\infty \frac{dt}{S(t)} \geq \int_0^\infty \frac{dt}{\sqrt{C_1 + C_2 t}} = \infty. \tag{71}$$

**Step 3. Convergence of $\bar{a}(t)$ to $a_K$.**

Define $V(t) := a_K - \bar{a}(t) \geq 0$. Since $\forall i, w_i(t) > \delta, \bar{a}(t) < a_K$, and $V(0) > 0$.

We claim that $\bar{a}(t)$ converges to $a_K$ as $t \to \infty$. There are two cases.

*Case A: $V(t)$ hits zero in finite time.* Assume there exists $T^* < \infty$ with $V(T^*) = 0$, i.e. $\bar{a}(T^*) = a_K$. Since $\bar{a}(t)$ is nondecreasing by (63) and bounded above by $a_K$, we must have

$$\bar{a}(t) = a_K \quad \text{for all } t \geq T^*. \tag{72}$$

Hence $\lim_{t\to\infty} \bar{a}(t) = a_K$ in this case.

*Case B: $V(t) > 0$ for all $t \geq 0$.* Then $1/V(t)$ is well-defined and differentiable for all $t$. From (63) we have

$$\frac{dV}{dt} = -\frac{d\bar{a}}{dt} = -\frac{1}{S(t)^2}\sum_{i\in A(t)}(a_i - \bar{a}(t))^2. \tag{73}$$

Whenever $V(t) > 0$, we also have $K \in A(t)$, because $a_K > \bar{a}(t)$ implies the raw derivative of $w_K$, $\dot{w}_K$, is positive and the clamping rule does not prevent it from increasing. Thus

$$\sum_{i\in A(t)}(a_i - \bar{a}(t))^2 \geq (a_K - \bar{a}(t))^2 = V(t)^2, \tag{74}$$

so

$$\frac{dV}{dt} \leq -\frac{V(t)^2}{S(t)^2}. \tag{75}$$

Rewriting,

$$\frac{d}{dt}\left(\frac{1}{V(t)}\right) = -\frac{1}{V(t)^2}\frac{dV}{dt} \geq \frac{1}{S(t)^2} \quad \text{for all } t \geq 0. \tag{76}$$

Integrating from $0$ to $T$ yields

$$\frac{1}{V(T)} - \frac{1}{V(0)} \geq \int_0^T \frac{dt}{S(t)^2}. \tag{77}$$

By (70), the integral on the right-hand side diverges as $T \to \infty$, hence

$$\frac{1}{V(T)} \xrightarrow[T \to \infty]{} \infty \quad \Rightarrow \quad V(T) \xrightarrow[T \to \infty]{} 0. \tag{78}$$

Thus $\bar{a}(t) \to a_K$ as $t \to \infty$ in this case as well.

Combining cases A and B, we conclude in all situations that

$$\lim_{t \to \infty} \bar{a}(t) = a_K. \tag{79}$$

**Step 4. Convergence of $p_K(t)$ to $1$ for all $\delta \geq 0$.**

For $i \neq K$, define $\Delta_i := a_K - a_i > 0$ and $\Delta_{\min} := \min_{i \neq K} \Delta_i > 0$. Then

$$V(t) = a_K - \bar{a}(t) = a_K - \sum_j p_j(t) a_j = \sum_j p_j(t)(a_K - a_j) = \sum_{j \neq K} p_j(t) \Delta_j. \tag{80}$$

Therefore

$$V(t) \geq \Delta_{\min} \sum_{j \neq K} p_j(t) = \Delta_{\min}\big(1 - p_K(t)\big), \tag{81}$$

so

$$1 - p_K(t) \leq \frac{V(t)}{\Delta_{\min}}. \tag{82}$$

Since $V(t) \to 0$, the right-hand side tends to $0$ and we obtain

$$\lim_{t \to \infty} p_K(t) = 1. \tag{83}$$

Because $\sum_i p_i(t) = 1$ and each $p_i(t) \geq 0$, this implies $p_i(t) \to 0$ for all $i \neq K$. This proves statement (1).

**Step 5. Finite-time convergence when $\delta = 0$.**

In this step we assume $\delta = 0$. Recall from (69) that there exist constants $C_1, C_2 > 0$ such that

$$S^2(t) \leq C_1 + C_2 t \quad \text{for all } t \geq 0, \tag{84}$$

hence

$$\frac{1}{S(t)} \geq \frac{1}{\sqrt{C_1 + C_2 t}}, \qquad \int_0^\infty \frac{dt}{S(t)} = \infty. \tag{85}$$

Recall also that $V(t) := a_K - \bar{a}(t) \geq 0$ and, from (76), as long as $V(t) > 0$, we have

$$\frac{d}{dt}\left(\frac{1}{V(t)}\right) \geq \frac{1}{S(t)^2} \geq \frac{1}{C_1 + C_2 t}, \tag{86}$$

With $V(t) > 0$, integrating yields

$$\frac{1}{V(t)} - \frac{1}{V(0)} \geq \frac{1}{C_2} \log\left(\frac{C_1 + C_2 t}{C_1}\right), \tag{87}$$

and hence

$$V(t) \leq \frac{1}{\dfrac{1}{V(0)} + \dfrac{1}{C_2} \log\left(\dfrac{C_1 + C_2 t}{C_1}\right)} \quad \text{whenever } V(t) > 0. \tag{88}$$

Fix an index $i \neq K$ and set $\Delta_i := a_K - a_i > 0$. We first identify an explicit time after which $\bar{a}(t)$ stays above the midpoint between $a_i$ and $a_K$. Define $T_{0,i}$ as any time satisfying

$$\bar{a}(T_{0,i}) \geq \frac{a_i + a_K}{2}, \tag{89}$$

which implies

$$V(T_{0,i}) = a_K - \bar{a}(T_{0,i}) \le a_K - \frac{a_i + a_K}{2} = \frac{a_K - a_i}{2} = \frac{\Delta_i}{2} \tag{90}$$

Plugging in (88),

$$\frac{1}{V(0)} + \frac{1}{C_2} \log\left(\frac{C_1 + C_2 T_{0,i}}{C_1}\right) \ge \frac{2}{\Delta_i}. \tag{91}$$

For instance, if $\frac{1}{V(0)} \ge \frac{2}{\Delta_i}$ one may take $T_{0,i} = 0$; otherwise any

$$T_{0,i} \ge \frac{C_1}{C_2}\left[\exp\left(C_2\left(\frac{2}{\Delta_i} - \frac{1}{V(0)}\right)\right) - 1\right] \tag{92}$$

satisfies (91).

Now consider the evolution of $w_i(t)$ for $t \ge T_{0,i}$. As long as $w_i(t) > 0$, the clamped dynamics coincide with the original ODE, so

$$\frac{dw_i}{dt}(t) = \frac{a_i - \bar{a}(t)}{S(t)}. \tag{93}$$

For $t \ge T_{0,i}$ with $w_i(t) > 0$ we have

$$a_i - \bar{a}(t) \le a_i - \frac{a_K + a_i}{2} = -\frac{\Delta_i}{2}, \tag{94}$$

hence

$$\frac{dw_i}{dt}(t) \le -\frac{\Delta_i}{2} \cdot \frac{1}{S(t)} \le -\frac{\Delta_i}{2} \cdot \frac{1}{\sqrt{C_1 + C_2 t}}. \tag{95}$$

Integrating from $T_{0,i}$ to $t$ (while $w_i > 0$) yields

$$w_i(t) \le w_i(T_{0,i}) - \frac{\Delta_i}{2} \int_{T_{0,i}}^{t} \frac{ds}{\sqrt{C_1 + C_2 s}}. \tag{96}$$

The integral can be computed explicitly:

$$\int_{T_{0,i}}^{t} \frac{ds}{\sqrt{C_1 + C_2 s}} = \frac{2}{C_2}\left(\sqrt{C_1 + C_2 t} - \sqrt{C_1 + C_2 T_{0,i}}\right), \tag{97}$$

so

$$w_i(t) \le w_i(T_{0,i}) - \frac{\Delta_i}{C_2}\left(\sqrt{C_1 + C_2 t} - \sqrt{C_1 + C_2 T_{0,i}}\right) \quad \text{for all } t \ge T_{0,i} \tag{98}$$

as long as $w_i(t) > 0$.

Define $T_i^*$ by requiring that the right-hand side of (98) is equal to zero:

$$w_i(T_{0,i}) - \frac{\Delta_i}{C_2}\left(\sqrt{C_1 + C_2 T_i^*} - \sqrt{C_1 + C_2 T_{0,i}}\right) = 0. \tag{99}$$

It follows that

$$T_i^* = \frac{1}{C_2}\left[\left(\sqrt{C_1 + C_2 T_{0,i}} + \frac{C_2}{\Delta_i} w_i(T_{0,i})\right)^2 - C_1\right], \tag{100}$$

where $C_1 = S^2(0)$ and $C_2 = 2N(a_{\max} - a_{\min}) > 0$.

Since $w_i(t) \ge 0$ for all $t$, the inequality (98) implies that $w_i(t)$ must hit 0 no later than $T_i^* < \infty$. In fact, once we cross the time $T_{0,i}$, $w_i(t)$ decays at least linearly in $\sqrt{t}$.

For $t \ge T_i^*$, we still have $\bar{a}(t) \ge (a_i + a_K)/2 > a_i$, so the $\dot{w}_i(t)$ is negative and the clamped dynamics enforce $w_i(t) \equiv 0$ for all $t \ge T_i^*$.

Since $i \ne K$ was arbitrary, every non-dominant coordinate $w_i$, $i \ne K$, hits zero in finite time and stays there. Define

$$T_* := \max_{i \ne K} T_i^* < \infty. \tag{101}$$

For all $t \geq T_*$ we then have

$$w_i(t) = 0 \quad (i \neq K), \qquad S(t) = w_K(t) > 0. \tag{102}$$

Thus $\bar{a}(t) = a_K$ for all $t \geq T_*$ and

$$\frac{dw_K}{dt}(t) = \frac{a_K - \bar{a}(t)}{S(t)} = 0 \quad \text{for } t \geq T_*, \tag{103}$$

so $w_K(t)$ is constant on $[T_*, \infty)$ and

$$p_K(t) = \frac{w_K(t)}{S(t)} = 1, \qquad p_i(t) = 0 \ (i \neq K), \quad \text{for all } t \geq T_*. \tag{104}$$

This establishes finite-time convergence when $\delta = 0$ and proves statement (2) of the theorem.

**Step 6. Polynomial-rate convergence when $\delta > 0$.**

Now assume $\delta > 0$. We first show that each non-dominant weight $w_i(t)$, $i \neq K$, hits the floor level $\delta$ in finite time and then stays there.

Fix $i \neq K$. As in Step 5, since $\bar{a}(t) \to a_K > a_i$ there exists $T_i^a$ such that

$$\bar{a}(t) \geq \frac{a_i + a_K}{2} \quad \text{for all } t \geq T_i^a. \tag{105}$$

For $t \geq T_i^a$ and while $w_i(t) > \delta$, the index $i$ is active and we have

$$\frac{dw_i}{dt}(t) = \frac{a_i - \bar{a}(t)}{S(t)} \leq -\frac{\Delta_i}{2} \cdot \frac{1}{S(t)}. \tag{106}$$

Arguing as before with (71), the integral $\int_{T_i^a}^{\infty} (1/S(t))\, dt$ diverges, so there exists $T_i^* \geq T_i^a$ such that $w_i(T_i) = \delta$. For $t \geq T_i^*$ we still have $\bar{a}(t) \geq (a_i + a_K)/2 > a_i$, so the raw derivative $\dot{w}_i$ at the boundary is negative and the clamped dynamics enforce $w_i(t) \equiv \delta$ for all $t \geq T_i$.

Let

$$T_0 := \max_{i \neq K} T_i^* < \infty. \tag{107}$$

For all $t \geq T_0$ we then have

$$w_i(t) = \delta \quad (i \neq K), \qquad w_K(t) > \delta, \qquad S(t) = w_K(t) + C_3, \tag{108}$$

where $C_3 := (N-1)\delta$ is a constant. The average $\bar{a}(t)$ is

$$\bar{a}(t) = \frac{w_K(t)a_K + \sum_{i \neq K} \delta a_i}{w_K(t) + C_3} = \frac{wa_K + \delta \sum_{i \neq K} a_i}{w + C_3}. \tag{109}$$

Hence

$$a_K - \bar{a}(t) = \frac{\delta\big[(N-1)a_K - \sum_{i \neq K} a_i\big]}{w_K(t) + C_3} = \frac{C_0}{w_K(t) + C_3}, \tag{110}$$

where

$$C_0 := \delta \sum_{i \neq K}(a_K - a_i) > 0. \tag{111}$$

Since $K$ is active for all $t \geq T_0$, its dynamics are

$$\frac{dw_K}{dt}(t) = \frac{a_K - \bar{a}(t)}{S(t)} = \frac{C_0}{(w_K(t) + C_3)^2}. \tag{112}$$

This scalar ODE can be integrated exactly:

$$(w_K(t) + C_3)^2 \frac{dw_K}{dt} = C_0, \tag{113}$$

so

$$\int_{w_K(T_0)}^{w_K(t)} (u + C_3)^2 \, du = C_0(t - T_0). \tag{114}$$

The antiderivative is $\frac{(u+C_3)^3}{3}$, hence

$$\frac{(w_K(t) + C_3)^3 - (w_K(T_0) + C_3)^3}{3} = C_0(t - T_0), \tag{115}$$

and therefore

$$w_K(t) + C_3 = \left( (w_K(T_0) + C_3)^3 + 3C_0(t - T_0) \right)^{1/3}. \tag{116}$$

Now, for $t \geq T_0$ we have

$$p_K(t) = \frac{w_K(t)}{S(t)} = \frac{w_K(t)}{w_K(t) + C_3}, \tag{117}$$

so

$$1 - p_K(t) = \frac{C_3}{w_K(t) + C_3} = \frac{C_3}{\left( (w_K(T_0) + C_3)^3 + 3C_0(t - T_0) \right)^{1/3}}. \tag{118}$$

This immediately implies $1 - p_K(t) \to 0$ and yields a quantitative rate. For all $t \geq T_0$,

$$(w_K(T_0) + C_3)^3 + 3C_0(t - T_0) \geq 3C_0(t - T_0), \tag{119}$$

so from (118)

$$1 - p_K(t) \leq \frac{C_3}{\left( 3C_0(t - T_0) \right)^{1/3}} = \frac{C_4}{(t - T_0)^{1/3}}, \tag{120}$$

where

$$C_4 := \frac{C_3}{(3C_0)^{1/3}} = \frac{(N - 1)\delta}{\left( 3\delta \sum_{i \neq K}(a_K - a_i) \right)^{1/3}}. \tag{121}$$

This proves the first inequality in statement (3).

Finally, fix $\epsilon \in (0, 1)$ and consider the exact formula (118). To ensure $1 - p_K(t) \leq \epsilon$, it suffices that

$$\frac{C_3}{w_K(t) + C_3} \leq \epsilon \Rightarrow w_K(t) + C_3 \geq \frac{C_3}{\epsilon}. \tag{122}$$

By (116) this holds whenever

$$(w_K(T_0) + C_3)^3 + 3C_0(t - T_0) \geq \left( \frac{C_3}{\epsilon} \right)^3. \tag{123}$$

Equivalently,

$$t - T_0 \geq \frac{1}{3C_0} \left( \frac{C_3^3}{\epsilon^3} - (w_K(T_0) + C_3)^3 \right). \tag{124}$$

For all sufficiently small $\epsilon$, the leading term $C_3^3/(3C_0\epsilon^3)$ dominates, so there exists a constant $C_5 > 0$ such that for all small $\epsilon$ one may choose

$$T_\epsilon := T_0 + \frac{C_3^3}{3C_0} \frac{1}{\epsilon^3} \leq T_0 + \frac{C_5}{\epsilon^3}, \tag{125}$$

and then $1 - p_K(t) \leq \epsilon$ for all $t \geq T_\epsilon$. This establishes the hitting-time bound in statement (3), completing the proof. $\square$

# C. Theorem 3.4 and Corollary 3.5 on the Sensitivity of $\Delta\phi$ to $w_K$

## C.1. Setup and Notations

For the convenience of the reader, we restate some notations used in this section. These notations are consistent throughout the main text and the appendix of the paper.

Let $w_i > 0$, $g_i \in \mathbb{R}^{d_g}$, $h_i \in \mathbb{R}^{d_h}$ for $i = 1, \ldots, N$. Restating (10):

$$\Delta\phi_t = -\frac{1}{S}\bar{\gamma}^\top \sum_{i=1}^{N} g_i(\theta_t)\left(h_i(\phi_t)^\top - \frac{w_i}{S}\sum_{j=1}^{N} h_j(\phi_t)^\top\right), \tag{10}$$

where $S = \sum_{i=1}^{N} w_i$ and $\bar{\gamma} = \frac{1}{N}\sum_{i=1}^{N} \gamma_i$.

We define a few new notations that help simplify the equation

$$p_i = \frac{w_i}{S}, \quad h_{\text{sum}} = \sum_{j=1}^{N} h_j, \quad a_i = \bar{\gamma}^\top g_i, \quad A = \bar{\gamma}^\top \sum_{i=1}^{N} g_i h_i^\top, \quad g_{\text{sum}} = \sum_{i=1}^{N} w_i g_i, \quad B = \bar{\gamma}^\top g_{\text{sum}} h_{\text{sum}}^\top. \tag{126}$$

So we have

$$\Delta\phi = \frac{1}{S}A - \frac{1}{S^2}B \in \mathbb{R}^{d_h}. \tag{127}$$

For a specific index $K$, the partial derivative of $\Delta\phi$ with respect to $w_K$ can be written as

$$\frac{\partial \Delta\phi}{\partial w_K} = \frac{1}{S^2}V_K, \qquad V_K = \sum_{i=1}^{N} a_i v_i, \tag{128}$$

where

$$v_i = (2p_i - \mathbb{1}_{\{i=K\}})h_{\text{sum}}^\top - h_i^\top \in \mathbb{R}^{d_h}. \tag{129}$$

## C.2. Assumptions

Next, we state the moderate assumptions necessary for the proof.

**Assumption C.1** (Across-index independence). The tuple of random variables $(x_i, g_i, w_i, h_i)$ are independent across the index $i$, but they are not necessarily independent within each tuple. $h_i$ are independent across $i$ whether conditioned on $w = [w_1, \ldots, w_N]$ or not. Similarly, $\gamma_i$ are independent across $i$.

**Assumption C.2** (The structure of $h_i$). Conditioning on $w = [w_1, \ldots, w_N]$ does not fully eliminate the uncertainty in $h_i$. This is reasonable despite the fact that $h_i = \frac{\partial w_i}{\partial \phi}$ because $w_i$ is a scalar and $h_i$ is a vector, and the function $s_\phi(x_i) = w_i$ could be many-to-one. Concretely, the conditional covariance satisfies

$$\text{Cov}(h_i \mid w) \succeq \Sigma_0, \tag{130}$$

for some fixed positive semi-definite matrix $\Sigma_0 \in \mathbb{R}^{d_h \times d_h}$, with $\text{tr}(\Sigma_0) > 0$. Further, there exists $M_h < \infty$ such that, for all $i$,

$$\mathbb{E}\big[\|h_i\|_2^4 \mid w\big] \leq M_h. \tag{131}$$

**Assumption C.3** (The structure of $a_i$). For each $i$, there exists a constant $\mu_i \in \mathbb{R}$ such that

$$\mathbb{E}[a_i \mid w, h] = \mu_i \tag{132}$$

where $w = [w_1, \ldots, w_N]$ and $h = [h_1, \ldots, h_N]$. Notice that $a_i = \bar{\gamma}^\top g_i$ depends on both the validation gradient $\bar{\gamma}$ and training data points $x_i$, so knowing $w$ and $h$ does not completely eliminate the uncertainty in $a_i$.

Further, the noise $\varepsilon_i := a_i - \mu_i$ satisfies

$$\mathbb{E}[\varepsilon_i^2 \mid w, h] \leq \sigma_u^2, \qquad \mathbb{E}[\varepsilon_i^4 \mid w, h] \leq M_u \tag{133}$$

for some finite constants $\sigma_u^2, M_u$, uniformly in $(w, h)$.

**Assumption C.4** (Spiky $a_i$, restatement of Assumption 3.3). There is a unique *dominant index* $K$ such that

$$\mu_K > \mu_i \quad \text{for all } i \neq K. \tag{134}$$

Define the "gap energy" $G_{\text{gap}}$ as

$$G_{\text{gap}}^2 := \frac{1}{N} \sum_{i \neq K} (\mu_K - \mu_i)^2 > 0. \tag{135}$$

### C.3. A Lemma

We introduce a lemma that is later used by Theorem 3.4.

**Lemma C.5.** *Fix an index $K \in \{1, \ldots, N\}$. Let $r = (r_1, \ldots, r_N)^\top \in \mathbb{R}^N$, $r_i \geq 0$, $r_K = 0$, $\sum_{i \neq K} r_i^2 = 1$. Let $p = (p_1, \ldots, p_N)^\top \in \mathbb{R}^N$ lie in the $(N-1)$-dimensional simplex, i.e. $p_i \geq 0, \sum_{i=1}^N p_i = 1$. Define*

$$f_{rp}(r, p) := \sum_{i=1}^N \left( r_i - 2 \sum_{j=1}^N p_j r_j \right)^2. \tag{136}$$

*The minimum of $f_{rp}(r, p)$ is*

$$\min_{(r,p)} f_{rp}(r, p) = \frac{1}{N}. \tag{137}$$

*attained at $r_K = 0$, $p_K = \frac{N+1}{2N}$, and for all $i \neq K$, $r_i = \frac{1}{\sqrt{N-1}}$, $p_i = \frac{1}{2N}$.*

*Proof.* Write

$$r_{\text{sum}} := \sum_{i=1}^N r_i, \qquad s_{\text{rp}} := p^\top r. \tag{138}$$

Expanding the square gives

$$f_{rp}(r, p) = \sum_{i=1}^N (r_i - 2s_{\text{rp}})^2 \tag{139}$$

$$= \sum_{i=1}^N r_i^2 - 4s_{\text{rp}} \sum_{i=1}^N r_i + 4N s_{\text{rp}}^2 \tag{140}$$

$$= 1 - 4s_{\text{rp}} r_{\text{sum}} + 4N s_{\text{rp}}^2, \tag{141}$$

using $\sum_{i=1}^N r_i^2 = \sum_{i \neq K} r_i^2 = 1$.

For fixed $r$, this is a quadratic function of $s_{\text{rp}}$:

$$g_r(s_{\text{rp}}) := 1 - 4s_{\text{rp}} r_{\text{sum}} + 4N s_{\text{rp}}^2, \tag{142}$$

with unconstrained minimizer at

$$s_{\text{rp}}^* = \frac{r_{\text{sum}}}{2N}. \tag{143}$$

Substituting this back in,

$$g_r(s_{\text{rp}}^*) = 1 - \frac{r_{\text{sum}}^2}{N}. \tag{144}$$

Next we bound $r_{\text{sum}}^2$ using Cauchy–Schwarz. Since $r_K = 0$ and $\sum_{i \neq K} r_i^2 = 1$, we have

$$r_{\text{sum}}^2 = \left( \sum_{i=1}^N r_i \right)^2 = \left( \sum_{i \neq K} r_i \right)^2 \leq (N-1) \sum_{i \neq K} r_i^2 = N - 1. \tag{145}$$

Thus

$$1 - \frac{r_{\text{sum}}^2}{N} \geq 1 - \frac{N-1}{N} = \frac{1}{N}, \tag{146}$$

and hence for all feasible $(r, p)$,

$$f_{rp}(r, p) \geq \frac{1}{N}. \tag{147}$$

It remains to show that this bound is tight. Define $r$ by

$$r_K = 0, \qquad r_i = \frac{1}{\sqrt{N-1}} \quad \text{for all } i \neq K. \tag{148}$$

We can verify that $\sum_{i \neq K} r_i^2 = (N-1) \cdot \frac{1}{N-1} = 1$, and

$$r_{\text{sum}} = \sum_{i=1}^{N} r_i = \sum_{i \neq K} r_i = (N-1) \cdot \frac{1}{\sqrt{N-1}} = \sqrt{N-1}. \tag{149}$$

Hence

$$g_r(s_{\text{rp}}^*) = 1 - \frac{r_{\text{sum}}^2}{N} = 1 - \frac{N-1}{N} = \frac{1}{N}. \tag{150}$$

We now construct $p$ so that $s_{\text{rp}} = s_{\text{rp}}^* = r_{\text{sum}}/(2N)$. Take

$$p_K = \frac{N+1}{2N}, \qquad p_i = \frac{1}{2N} \quad \text{for all } i \neq K. \tag{151}$$

We can verify that $p_i \geq 0$, $\sum_i p_i = 1$, and

$$s_{\text{rp}} = \sum_{i=1}^{N} p_i r_i = \sum_{i \neq K} \frac{1}{2N} \cdot \frac{1}{\sqrt{N-1}} = \frac{N-1}{2N\sqrt{N-1}} = \frac{\sqrt{N-1}}{2N} = s_{\text{rp}}^*. \tag{152}$$

Therefore, for this choice of $(r, p)$,

$$f_{rp}(r, p) = g_r(s_{\text{rp}}^*) = \frac{1}{N}, \tag{153}$$

showing that the lower bound is attained. Hence

$$\min_{(r,p)} f_{rp}(r, p) = \frac{1}{N}, \tag{154}$$

as claimed. □

### C.4. Theorem 3.4 and Its Proof

**Theorem 3.4** (Gradient lower bound in a spiky regime, restated). *Define the gap energy $G_{gap}$ as $G_{gap} := \sqrt{\frac{1}{N} \sum_{i \neq K} (\mu_K - \mu_i)^2}$. Given Assumptions C.1 to C.4, we have*

$$P\left( \left\| \frac{\partial \Delta \phi}{\partial w_K} \right\|_2 \geq \frac{\sqrt{\alpha} G_{gap}}{S^2} \right) \geq \frac{(\tau_h - \alpha)^2 G_{gap}^4}{K_1 N^9 G_{gap}^4 + K_2 N^8}, \tag{155}$$

*where $\alpha$ is an arbitrary value in $(0, \tau_h)$; $\tau_h$, $K_1$, and $K_2$ are positive constants that depend only on the distribution of $(w_i, g_i, h_i)$ and not on $N$ or $G_{gap}$.*

*Proof.* We first compute $V_K$ explicitly and then lower bound its typical value.

**Step 1: Expression for $V_K$ and its conditional mean.** By differentiating $\Delta\phi = S^{-1}A - S^{-2}B$ with respect to $w_K$ and using the identities

$$\frac{\partial S^{-1}}{\partial w_K} = -\frac{1}{S^2}, \qquad \frac{\partial S^{-2}}{\partial w_K} = -\frac{2}{S^3}, \qquad \frac{\partial B}{\partial w_K} = \bar{\gamma}^\top g_K h_{\text{sum}}^\top, \tag{156}$$

one checks that

$$\frac{\partial \Delta\phi}{\partial w_K} = -\frac{A}{S^2} - \frac{1}{S^2}\bar{\gamma}^\top g_K h_{\text{sum}}^\top + \frac{2}{S^3}B = \frac{1}{S^2}\Big(-A + \bar{\gamma}^\top(2\bar{g} - g_K)h_{\text{sum}}^\top\Big), \tag{157}$$

where $\bar{g} = \sum_i p_i g_i$. Writing $a_i = \bar{\gamma}^\top g_i$ and $h_{\text{sum}} = \sum_j h_j$, one can rewrite this as

$$V_K = -A + \bar{\gamma}^\top(2\bar{g} - g_K)h_{\text{sum}}^\top = \sum_{i=1}^N a_i v_i, \tag{158}$$

with

$$v_i = (2p_i - \mathbb{1}_{\{i=K\}})h_{\text{sum}}^\top - h_i^\top. \tag{159}$$

Define mean conditioned on

$$m(w,h) := \mathbb{E}[V_K \mid w, h]. \tag{160}$$

By Assumption C.4, $\mathbb{E}[a_i \mid w, h] = \mu_i$, and $v_i$ is a deterministic function of $(w, h)$, hence

$$m(w,h) = \sum_{i=1}^N \mu_i v_i. \tag{161}$$

Moreover,

$$\mathbb{E}\big[\|V_K\|_2^2 \mid w, h\big] = \|m(w,h)\|_2^2 + \text{tr}\big(\text{Cov}(V_K \mid w, h)\big) \geq \|m(w,h)\|_2^2. \tag{162}$$

**Step 2: Writing $m(w,h)$ as a linear combination of $h_i$.** Using $h_{\text{sum}}^\top = \sum_j h_j^\top$, we have

$$m(w,h) = \sum_i \mu_i\big[(2p_i - \mathbb{1}_{\{i=K\}})h_{\text{sum}}^\top - h_i^\top\big] = \alpha_0(w)\sum_i h_i^\top - \sum_i \mu_i h_i^\top, \tag{163}$$

where

$$\alpha_0(w) := \sum_i \mu_i(2p_i - \mathbb{1}_{\{i=K\}}) = 2\sum_i p_i \mu_i - \mu_K. \tag{164}$$

Thus there are scalar coefficients

$$c_i(w) := \alpha_0(w) - \mu_i \tag{165}$$

such that

$$m(w,h) = \sum_{i=1}^N c_i(w)\, h_i. \tag{166}$$

**Step 3: Conditional second moment of $m$ given $w$.** Assumptions C.1 and C.2 tell us that given $w$, $h_i$ are conditionally independent and

$$\text{Cov}(h_i \mid w) \succeq \Sigma_0. \tag{167}$$

Hence

$$\text{Cov}(m \mid w) = \sum_i c_i(w)^2\, \text{Cov}(h_i \mid w), \tag{168}$$

so that

$$\mathbb{E}_h\big[\|m(w,h)\|_2^2 \mid w\big] = \text{tr}\big(\text{Cov}(m \mid w)\big) + \|\mathbb{E}_h(m \mid w)\|_2^2 \geq \sum_i c_i(w)^2\, \text{tr}(\text{Cov}(h_i \mid w)). \tag{169}$$

By $\text{Cov}(h_i \mid w) \succeq \Sigma_0$ we have $\text{tr}(\text{Cov}(h_i \mid w)) \geq \tau_h := \text{tr}(\Sigma_0)$, hence

$$\mathbb{E}_h\big[\|m(w,h)\|_2^2 \mid w\big] \geq \tau_h \sum_i c_i(w)^2. \tag{170}$$

**Step 4: Relating $\sum_i c_i(w)^2$ to the gap energy.** Define

$$G_{\text{gap}}^2 \;:=\; \frac{1}{N} \sum_{i \neq K} (\mu_K - \mu_i)^2 \;>\; 0. \tag{171}$$

We claim that for all $w$,

$$\sum_{i=1}^{N} c_i(w)^2 \;\geq\; G_{\text{gap}}^2. \tag{172}$$

To see this, fix the vector $(\mu_i)$ and consider the normalized gaps $r_i$, defined as

$$r_i := (\mu_K - \mu_i)/(\sqrt{N} G_{\text{gap}}) \text{ for } i \neq K, \text{ and } r_K = 0, \tag{173}$$

so that $\sum_{i \neq K} r_i^2 = 1$. We can show that

$$\sum_i c_i(w)^2 \;=\; N G_{\text{gap}}^2 f_{rp}(r, p), \quad f_{rp}(r, p) := \sum_{i=1}^{N} \left( r_i - 2 \sum_{j=1}^{N} p_j r_j \right)^2. \tag{174}$$

The following steps show how (174) is derived. For all $i$ we can write

$$\mu_i = \mu_K - \sqrt{N} G_{\text{gap}} \, r_i. \tag{175}$$

Hence,

$$\sum_i p_i \mu_i = \sum_i p_i (\mu_K - \sqrt{N} G_{\text{gap}} r_i) = \mu_K - \sqrt{N} G_{\text{gap}} \sum_i p_i r_i, \tag{176}$$

$$\alpha_0(w) = 2\big(\mu_K - \sqrt{N} G_{\text{gap}} \sum_i p_i r_i\big) - \mu_K = \mu_K - 2\sqrt{N} G_{\text{gap}} \sum_i p_i r_i. \tag{177}$$

Therefore

$$c_i(w) := \alpha_0(w) - \mu_i \tag{178}$$

$$= \big(\mu_K - 2\sqrt{N} G_{\text{gap}} \sum_j p_j r_j\big) - \big(\mu_K - \sqrt{N} G_{\text{gap}} r_i\big) \tag{179}$$

$$= \sqrt{N} G_{\text{gap}} \Big( r_i - 2 \sum_j p_j r_j \Big), \tag{180}$$

and the exact form of $f_{rp}$ follows.

By Lemma C.5, under the constraints $r_i \geq 0$, $\sum_{i \neq K} r_i^2 = 1$, $p_i \geq 0$, $\sum_i p_i = 1$,

$$\min f_{rp}(r, p) = \frac{1}{N}. \tag{181}$$

Therefore (172) holds for all $w$.

Combining (170) and (172), we obtain a bound independent of $w$ and $h$:

$$\mathbb{E}_h\big[\|m(w, h)\|_2^2 \mid w\big] \;\geq\; \tau_h G_{\text{gap}}^2. \tag{182}$$

**Step 5: Lower bound for $\mathbb{E}[\|V_K\|_2^2]$.** Set

$$Z \;:=\; \|V_K\|_2^2. \tag{183}$$

Recall that $m(w, h) = \mathbb{E}[V_K \mid w, h]$, and the second moment is lower bounded by the square of the mean,

$$\mathbb{E}[Z] = \mathbb{E}\big[\|V_K\|_2^2\big] = \mathbb{E}_{w,h}\big[\mathbb{E}[\|V_K\|_2^2 \mid w, h]\big] \;\geq\; \mathbb{E}_{w,h}\big[\|m(w, h)\|_2^2\big] \tag{184}$$

Using the bound (182),

$$\mathbb{E}_{w,h}\big[\|m(w, h)\|_2^2\big] = \mathbb{E}_w\Big[\mathbb{E}_h[\|m(w, h)\|_2^2 \mid w]\Big] \;\geq\; \tau_h G_{\text{gap}}^2. \tag{185}$$

Hence

$$\mathbb{E}[Z] \;\geq\; \tau_h G_{\text{gap}}^2. \tag{186}$$

**Step 6: Upper bound for $\mathbb{E}[Z^2]$ with explicit $N$-dependence.** Recall that

$$Z \; := \; \|V_K\|_2^2, \qquad V_K = \sum_{i=1}^{N} a_i v_i, \qquad a_i = \mu_i + \varepsilon_i \in \mathbb{R}. \tag{187}$$

*Step 6(a). Re-centering the means.* Define the average

$$\bar{\mu} := \frac{1}{N} \sum_{i=1}^{N} \mu_i, \qquad \delta_i := \mu_i - \bar{\mu}, \tag{188}$$

so that $\sum_{i=1}^{N} \delta_i = 0$ and $a_i = \bar{\mu} + \delta_i + \varepsilon_i$. Using the definitions $v_i = (2p_i - \mathbb{1}_{\{i=K\}})h_{\text{sum}} - h_i$, $h_{\text{sum}} = \sum_{j=1}^{N} h_j$, we obtain

$$\sum_{i=1}^{N} v_i = \sum_{i=1}^{N} (2p_i - \mathbb{1}_{\{i=K\}})h_{\text{sum}} - \sum_{i=1}^{N} h_i \tag{189}$$

$$= (2\sum_{i=1}^{N} p_i - 1)h_{\text{sum}} - h_{\text{sum}} = (2-1)h_{\text{sum}} - h_{\text{sum}} = 0. \tag{190}$$

Therefore

$$V_K = \sum_{i=1}^{N} a_i v_i = \sum_{i=1}^{N} (\bar{\mu} + \delta_i + \varepsilon_i)v_i = \sum_{i=1}^{N} \delta_i v_i + \sum_{i=1}^{N} \varepsilon_i v_i. \tag{191}$$

where we used $\sum_{i=1}^{N} v_i = 0$.

Using the inequality $(x+y)^4 \leq 8(x^4 + y^4)$ for all $x, y \in \mathbb{R}$, applied to norms, we obtain

$$Z^2 = \|V_K\|_2^4 \; \leq \; 8\left\|\sum_{i=1}^{N} \delta_i v_i\right\|_2^4 + 8\left\|\sum_{i=1}^{N} \varepsilon_i v_i\right\|_2^4. \tag{192}$$

*Step 6(b). A bound on $\mathbb{E}\|v_i\|_2^4$.* Recall

$$v_i = (2p_i - \mathbb{1}_{\{i=K\}})\, h_{\text{sum}} - h_i, \qquad h_{\text{sum}} = \sum_{j=1}^{N} h_j. \tag{193}$$

Since $|2p_i - \mathbb{1}_{\{i=K\}}| \leq 2$, there exists a constant $C_3 > 0$ such that

$$\|v_i\|_2 \; \leq \; C_3\big(\|h_{\text{sum}}\|_2 + \|h_i\|_2\big), \tag{194}$$

and hence, by $(a+b)^4 \leq 8(a^4 + b^4)$,

$$\|v_i\|_2^4 \; \leq \; C_4\Big(\|h_{\text{sum}}\|_2^4 + \|h_i\|_2^4\Big) \tag{195}$$

for some constant $C_4 > 0$ independent of $N$.

Write $h_j = m + z_j$, where $m := \mathbb{E}[h_1]$ and $\mathbb{E}[z_j] = 0$. Then

$$h_{\text{sum}} = \sum_{j=1}^{N} h_j = Nm + \sum_{j=1}^{N} z_j, \tag{196}$$

so

$$\|h_{\text{sum}}\|_2^4 \leq 8\, \|Nm\|_2^4 + 8\left\|\sum_{j=1}^{N} z_j\right\|_2^4. \tag{197}$$

Standard fourth-moment bounds for sums of independent mean-zero random vectors imply that there is a constant $C_5 > 0$ (depending only on the one-sample distribution of $h_i$ and the dimension $d_h$, but not on $N$) such that

$$\mathbb{E}\Big\| \sum_{j=1}^{N} z_j \Big\|_2^4 \leq C_5 N^2. \tag{198}$$

Therefore,

$$\mathbb{E}\|h_{\text{sum}}\|_2^4 \leq 8N^4\|m\|_2^4 + 8C_5 N^2. \tag{199}$$

Taking expectations in (195) and using Assumption C.2, which guarantees $\mathbb{E}\|h_i\|_2^4 \leq M_h$, we obtain

$$\mathbb{E}\|v_i\|_2^4 \leq C_4\Big(\mathbb{E}\|h_{\text{sum}}\|_2^4 + \mathbb{E}\|h_i\|_2^4\Big) \leq C_4\big(8N^4\|m\|_2^4 + 8C_5 N^2 + M_h\big) \leq C_v N^4, \tag{200}$$

for some constant $C_v > 0$ depending only on the one-sample distribution of $h_i$, but not on $N$ or $G_{\text{gap}}$.

*Step 6(c). Bounding the signal term* $\mathbb{E}\big\| \sum_{i=1}^{N} \delta_i v_i \big\|_2^4$. For arbitrary vectors $\beta_1, \ldots, \beta_N$,

$$\Big\| \sum_{i=1}^{N} \beta_i \Big\|_2^4 \leq \Big( \sum_{i=1}^{N} \|\beta_i\|_2 \Big)^4 \leq N^3 \sum_{i=1}^{N} \|\beta_i\|_2^4. \tag{201}$$

Applying this with $\beta_i := \delta_i v_i$ gives

$$\Big\| \sum_{i=1}^{N} \delta_i v_i \Big\|_2^4 \leq N^3 \sum_{i=1}^{N} |\delta_i|^4 \|v_i\|_2^4. \tag{202}$$

Taking expectations and using the bound on $\mathbb{E}\|v_i\|_2^4$ from (200),

$$\mathbb{E}\Big\| \sum_{i=1}^{N} \delta_i v_i \Big\|_2^4 \leq N^3 \sum_{i=1}^{N} |\delta_i|^4 \, \mathbb{E}\|v_i\|_2^4 \leq N^3 \sum_{i=1}^{N} |\delta_i|^4 \, C_v N^4 = C_v N^7 \sum_{i=1}^{N} \delta_i^4. \tag{203}$$

Next, we relate the centered means $(\delta_i)$ to the gap energy $G_{\text{gap}} = \sqrt{\frac{1}{N} \sum_{i \neq K} (\mu_K - \mu_i)^2}$. Recall $\mu_i = \bar{\mu} + \delta_i$. Then

$$\mu_K - \mu_i = (\bar{\mu} + \delta_K) - (\bar{\mu} + \delta_i) = \delta_K - \delta_i, \tag{204}$$

so

$$G_{\text{gap}}^2 = \frac{1}{N} \sum_{i \neq K} (\delta_K - \delta_i)^2 = \frac{1}{N} \sum_{i \neq K} \big(\delta_K^2 - 2\delta_K\delta_i + \delta_i^2\big)$$

$$= \frac{1}{N} \left( (N-1)\delta_K^2 - 2\delta_K \sum_{i \neq K} \delta_i + \sum_{i \neq K} \delta_i^2 \right).$$

Using $\sum_{i=1}^{N} \delta_i = 0$ gives $\sum_{i \neq K} \delta_i = -\delta_K$, hence

$$N G_{\text{gap}}^2 = (N-1)\delta_K^2 + 2\delta_K^2 + \sum_{i \neq K} \delta_i^2 = (N+1)\delta_K^2 + \sum_{i \neq K} \delta_i^2 \geq \sum_{i=1}^{N} \delta_i^2. \tag{205}$$

Thus

$$\sum_{i=1}^{N} \delta_i^2 \leq N G_{\text{gap}}^2. \tag{206}$$

Since all $\delta_i^2 \geq 0$, we have

$$\sum_{i=1}^{N} \delta_i^4 \leq \Big( \sum_{i=1}^{N} \delta_i^2 \Big)^2 \leq N^2 G_{\text{gap}}^4. \tag{207}$$

Substituting into (203) yields

$$\mathbb{E}\Big\|\sum_{i=1}^{N}\delta_i v_i\Big\|_2^4 \;\leq\; C_v N^9 G_{\text{gap}}^4. \tag{208}$$

*Step 6(d). Bounding the noise term* $\mathbb{E}\big\|\sum_{i=1}^{N}\varepsilon_i v_i\big\|_2^4$. Applying the inequality (201) with $\beta_i := \varepsilon_i v_i$,

$$\Big\|\sum_{i=1}^{N}\varepsilon_i v_i\Big\|_2^4 \;\leq\; N^3 \sum_{i=1}^{N}|\varepsilon_i|^4 \,\|v_i\|_2^4. \tag{209}$$

Taking expectations,

$$\begin{aligned}
\mathbb{E}\Big\|\sum_{i=1}^{N}\varepsilon_i v_i\Big\|_2^4 &= \mathbb{E}\Big[N^3 \sum_{i=1}^{N}|\varepsilon_i|^4 \,\|v_i\|_2^4\Big] \\
&\leq N^3 \sum_{i=1}^{N}\mathbb{E}\Big[\|v_i\|_2^4 \,\mathbb{E}\big[|\varepsilon_i|^4 \mid w, h\big]\Big] \\
&\leq N^3 \sum_{i=1}^{N}\mathbb{E}\Big[\|v_i\|_2^4 M_u\Big] \quad \text{(by Assumption C.3)} \\
&\leq N^3 \sum_{i=1}^{N} M_u \,\mathbb{E}\|v_i\|_2^4 \;\leq\; N^3 \sum_{i=1}^{N} M_u \cdot C_v N^4 \\
&= C_v M_u N^8.
\end{aligned} \tag{210}$$

*Step 6(e). Putting the bounds together.* Combining (192), (208), and (210), we obtain

$$\mathbb{E}[Z^2] = \mathbb{E}\|V_K\|_2^4 \;\leq\; 8\,\mathbb{E}\Big\|\sum_{i=1}^{N}\delta_i v_i\Big\|_2^4 + 8\,\mathbb{E}\Big\|\sum_{i=1}^{N}\varepsilon_i v_i\Big\|_2^4 \;\leq\; 8 C_v N^9 G_{\text{gap}}^4 + 8 C_v M_u N^8. \tag{211}$$

Define

$$K_1 := 8 C_v, \qquad K_2 := 8 C_v M_u. \tag{212}$$

Thus

$$\mathbb{E}[Z^2] \;\leq\; K_1 N^9 G_{\text{gap}}^4 + K_2 N^8. \tag{213}$$

**Step 7: Paley–Zygmund inequality.** Recall from Step 5 that

$$\mathbb{E}[Z] = \mathbb{E}\big[\|V_K\|_2^2\big] \;\geq\; \tau_h G_{\text{gap}}^2. \tag{186}$$

We now apply the Paley–Zygmund inequality to the nonnegative random variable $Z$. For any $\lambda \in (0, \mathbb{E}[Z])$,

$$\mathbb{P}(Z \geq \lambda) \;\geq\; \frac{(\mathbb{E}[Z] - \lambda)^2}{\mathbb{E}[Z^2]}. \tag{214}$$

We choose $\lambda$ proportional to $G_{\text{gap}}^2$:

$$\lambda := \alpha\, G_{\text{gap}}^2 \tag{215}$$

for some $0 < \alpha < \tau_h/N$. Then $0 < \lambda < \mathbb{E}[Z]$ and

$$\mathbb{E}[Z] - \lambda \;\geq\; (\tau_h - \alpha) G_{\text{gap}}^2. \tag{216}$$

Using the lower bound on $\mathbb{E}[Z]$ and the upper bound (213) on $\mathbb{E}[Z^2]$, we obtain

$$\mathbb{P}\big(Z \geq \alpha G_{\text{gap}}^2\big) \;\geq\; \frac{(\tau_h - \alpha)^2 G_{\text{gap}}^4}{K_1 N^9 G_{\text{gap}}^4 + K_2 N^8}. \tag{217}$$

Recall that

$$\frac{\partial \Delta \phi}{\partial w_K} \;=\; \frac{1}{S^2}\, V_K, \qquad S = \sum_{i=1}^{N} w_i, \tag{218}$$

so on the event $\{Z \geq \lambda\}$ (equivalently $\{\|V_K\|_2 \geq \sqrt{\lambda}\}$) we have

$$\left\| \frac{\partial \Delta \phi}{\partial w_K} \right\|_2 \;=\; \frac{\|V_K\|_2}{S^2} \;\geq\; \frac{\sqrt{\lambda}}{S^2} \;=\; \frac{\sqrt{\alpha}\, G_{\mathrm{gap}}}{S^2}. \tag{219}$$

Therefore,

$$\mathbb{P}\!\left( \left\| \frac{\partial \Delta \phi}{\partial w_K} \right\|_2 \;\geq\; \frac{\sqrt{\alpha}\, G_{\mathrm{gap}}}{S^2} \right) \;\geq\; \frac{(\tau_h - \alpha)^2 G_{\mathrm{gap}}^4}{K_1 N^9 G_{\mathrm{gap}}^4 + K_2 N^8}, \tag{220}$$

for any $\alpha \in (0, \tau_h)$, where $\tau_h$, $K_1$, and $K_2$ are positive constants depending only on the distribution of $(w_i, g_i, h_i)$ and not on $N$ or $G_{\mathrm{gap}}$.

$\square$

## C.5. Corollary 3.5: A Local Lower Bound on the Lipschitz-like Ratio $\frac{\|\widetilde{\widetilde{\Delta\phi}} - \Delta\phi\|_2}{|\widetilde{w}_K - w_K|}$

In this subsection, we consider the effect of resampling the dominant data point $(x_K, y_K)$, which enters the calculation through the following: $w_i = s_\phi(x_i)$, $h_i = \frac{\partial s_\phi(x_i)}{\partial \phi}$ and $g_i = \frac{\partial f(x_i)}{\partial \theta}$. Hence, this is equivalent to resampling the tuple $(g_K, w_K, h_K)$. More formally, we consider the following setting:

**Definition C.7** (Resampled configuration). Let $(g_i, w_i, h_i)_{i=1}^{N}$ be the original sample and let $\Delta\phi$ be defined as in (10). Let $(g_K', w_K', h_K')$ be an i.i.d copy of $(g_K, w_K, h_K)$, and set

$$(g_i^\dagger, w_i^\dagger, h_i^\dagger) := \begin{cases} (g_i, w_i, h_i), & i \neq K, \\ (g_K', w_K', h_K'), & i = K. \end{cases} \tag{221}$$

Let $\widetilde{\widetilde{\Delta\phi}}$ be the quantity obtained by applying (10) to the resampled tuples $(g_i^\dagger, w_i^\dagger, h_i^\dagger)_{i=1}^{N}$, and let $\delta w := w_K' - w_K$.

We also need one more mild assumption:

**Assumption C.8** (Non-degenerate support of $(g_i, w_i, h_i)$). In addition to Assumption C.1, assume that the law of $(g_i, w_i, h_i) \in \mathbb{R}^{d_g} \times (0, \infty) \times \mathbb{R}^{d_h}$ is absolutely continuous with respect to Lebesgue measure. That is, there exists a Borel-measurable function $f : \mathbb{R}^{d_g + 1 + d_h} \to [0, \infty)$ such that for every Borel set $A$,

$$P\big((g_i, w_i, h_i) \in A\big) = \int_A f(z)\, dz. \tag{222}$$

Moreover, there exists a nonempty open set $D \subset \mathbb{R}^{d_g + 1 + d_h}$ such that $f(z) > 0$ for all $z \in D$. With the event $E_{\mathrm{grad}} := \left\{ \left\| \frac{\partial \Delta\phi}{\partial w_K} \right\|_2 \geq \frac{\sqrt{\alpha} G_{\mathrm{gap}}}{S^2} \right\}$, $P(E_{\mathrm{grad}} \cap D^N) > 0$.

**Corollary 3.5** (Local Lipschitz-type lower bound under resampling, restated). *Let $K \in \{1, \dots, N\}$ be the dominant index. Suppose Assumptions C.1, C.2, C.3, C.4, and C.8 hold. Let $(g_i^\dagger, w_i^\dagger, h_i^\dagger)$ and the associated $\widetilde{\widetilde{\Delta\phi}}$ be defined as in Definition C.7. Then there exists a constant $K_3 > 0$, depending only on the distribution and dimensions of $(g_i, w_i, h_i)$ and the validation gradient $\bar{\gamma}$ (but not on $N$ or $G_{\mathrm{gap}}$), such that for every $\alpha \in (0, \tau_h)$,*

$$P\!\left( \frac{\|\widetilde{\widetilde{\Delta\phi}} - \Delta\phi\|_2}{|\delta w|} \;\geq\; K_3 \frac{\sqrt{\alpha}\, G_{\mathrm{gap}}}{S^2} \right) \;>\; 0. \tag{223}$$

*Proof.* The proof has three parts. We first show that there exists a deterministic configuration where the directional derivative of $\Delta\phi$ with respect to $w_K$ is nonzero. We then use a Taylor expansion in $w_K$ to produce a finite-difference lower bound at that configuration. Finally, we invoke absolute continuity of the joint distribution and continuity of $\Delta\phi$ to lift this to an event of strictly positive probability under the resampling scheme.

**Step 1: Existence of a configuration with large** $\partial \Delta\phi/\partial w_K$. This step is mostly just bookkeeping required by mathematical rigor. By Theorem 3.4, under Assumptions C.1, C.2, C.3, and C.4, there exist constants $\tau_h, K_1, K_2 > 0$ (independent of $N$ and $G_{\text{gap}}$) such that for all $\alpha \in (0, \tau_h)$,

$$P\left( \left\| \frac{\partial \Delta\phi}{\partial w_K} \right\|_2 \geq \frac{\sqrt{\alpha} G_{\text{gap}}}{S^2} \right) \geq \frac{(\tau_h - \alpha)^2 G_{\text{gap}}^4}{K_1 N^9 G_{\text{gap}}^4 + K_2 N^8} > 0. \tag{224}$$

Define the gradient event

$$E_{\text{grad}} := \left\{ \left\| \frac{\partial \Delta\phi}{\partial w_K} \right\|_2 \geq \frac{\sqrt{\alpha} G_{\text{gap}}}{S^2} \right\}. \tag{225}$$

Then $P(E_{\text{grad}}) > 0$.

Let $Z$ denote the full configuration

$$Z := (z_1, \ldots, z_N) \quad \text{with} \quad z_i := (g_i, w_i, h_i) \in \mathbb{R}^{d_g + 1 + d_h}, \tag{226}$$

viewed as a random vector in $\left( \mathbb{R}^{d_g} \times (0, \infty) \times \mathbb{R}^{d_h} \right)^N$. By Assumption C.8, each $z_i$ has density $f > 0$ on the open set $D \subset \mathbb{R}^{d_g + 1 + d_h}$, so $Z$ has joint density

$$f_Z(Z) = \prod_{i=1}^{N} f(z_i), \tag{227}$$

which is strictly positive on $D^N$. In particular, the law of $Z$ is supported on $D^N$ and $f_Z(Z) > 0$ for all $Z \in D^N$.

Since $P(E_{\text{grad}} \cap D^N) > 0$ and $Z$ has density $f_Z$, the integral

$$\int_{E_{\text{grad}} \cap D^N} f_Z(Z) \, dZ > 0, \tag{228}$$

so there exists at least one point

$$Z^\star = (z_1^\star, \ldots, z_N^\star) = (g_1^\star, w_1^\star, h_1^\star, \ldots, g_N^\star, w_N^\star, h_N^\star) \in E_{\text{grad}} \cap D^N \tag{229}$$

such that $f_Z(Z^\star) > 0$. In particular, each triple $(g_i^\star, w_i^\star, h_i^\star) \in D$, and the corresponding sum of weights

$$S^\star := \sum_{i=1}^{N} w_i^\star > 0. \tag{230}$$

At this fixed configuration $Z^\star$, define

$$G^\star := \left\| \frac{\partial \Delta\phi}{\partial w_K}(Z^\star) \right\|_2. \tag{231}$$

From $Z^\star \in E_{\text{grad}}$, we have

$$G^\star \geq \frac{\sqrt{\alpha} G_{\text{gap}}}{(S^\star)^2}, \tag{232}$$

where $S^\star = \sum_{i=1}^{N} w_i^\star$.

**Step 2: Local finite-difference lower bound around** $Z^\star$. We would like to apply a Taylor-style argument, but $\frac{\partial \Delta\phi}{\partial w_K}$ is a vector, which creates some difficulties. To work around that, we single out one component, indexed by $j^\star$, from the vector and focus the Taylor argument there.

We define $j^\star$ as an index in the vector $\frac{\partial \Delta\phi}{\partial w_K}(Z^\star)$ that takes on the maximum absolute value.

$$j^\star := \arg\max_{1 \leq j \leq d_h} \left| \left[ \frac{\partial \Delta\phi}{\partial w_K}(Z^\star) \right]_j \right| \tag{233}$$

We now treat $[\Delta\phi]_{j^\star}$ as a scalar function of $w_K$ while keeping all other coordinates fixed at $Z^\star$.

Define the scalar function

$$\psi(w) := [\Delta\phi(Z(w))]_{j^\star}, \tag{234}$$

where $Z(w)$ denotes the configuration obtained from $Z^\star$ by replacing $w_K^\star$ with $w$:

$$Z(w) := \left(g_1^\star, w_1^\star, h_1^\star, \ldots, g_K^\star, w, h_K^\star, \ldots, g_N^\star, w_N^\star, h_N^\star\right). \tag{235}$$

Because $w_i > 0$ for all $i$, the sum

$$S(w) = \sum_{i \neq K} w_i^\star + w \tag{236}$$

is strictly positive for all $w$ in a neighborhood of $w_K^\star$. The map $Z \mapsto \Delta\phi(Z)$ is rational in the weights $(w_1, \ldots, w_N)$ with denominators $S$ and $S^2$, so it is twice continuously differentiable ($C^2$) in any region where $S > 0$. Therefore $\psi$ is $C^2$ in a neighborhood of $w_K^\star$.

Let

$$\psi'(w_K^\star) := \left[\frac{\partial\Delta\phi}{\partial w_K}(Z^\star)\right]_{j^\star}. \tag{237}$$

Then from the preceding bound (232) we have

$$|\psi'(w_K^\star)| \geq \frac{\sqrt{\alpha}}{\sqrt{d_h}} \frac{G_{\mathrm{gap}}}{(S^\star)^2}. \tag{238}$$

By $C^2$-smoothness, there exists $r_1 > 0$ such that $[w_K^\star - r_1, w_K^\star + r_1] \subset (0, \infty)$ and $\psi''$ is continuous on this interval. Define

$$L_1 := \sup_{|w - w_K^\star| \leq r_1} |\psi''(w)| < \infty. \tag{239}$$

Now set

$$r_2 := \min\left\{r_1, \frac{|\psi'(w_K^\star)|}{4L_1 + 1}\right\} > 0. \tag{240}$$

For any $w$ with $0 < |w - w_K^\star| \leq r_2$, Taylor's theorem gives

$$\psi(w) - \psi(w_K^\star) = \psi'(w_K^\star)(w - w_K^\star) + \tfrac{1}{2}\psi''(\xi)(w - w_K^\star)^2 \tag{241}$$

for some $\xi$ between $w_K^\star$ and $w$. Dividing by $|w - w_K^\star|$,

$$\frac{|\psi(w) - \psi(w_K^\star)|}{|w - w_K^\star|} \geq |\psi'(w_K^\star)| - \frac{1}{2}|\psi''(\xi)|\,|w - w_K^\star|. \tag{242}$$

Using $|\psi''(\xi)| \leq L_1$ and $|w - w_K^\star| \leq r_2$,

$$\frac{1}{2}|\psi''(\xi)|\,|w - w_K^\star| \leq \frac{1}{2}L_1 r_2 \leq \frac{1}{2}L_1 \frac{|\psi'(w_K^\star)|}{4L_1 + 1} \leq \frac{1}{4}|\psi'(w_K^\star)|. \tag{243}$$

Hence, for all $w$ with $0 < |w - w_K^\star| \leq r_2$,

$$\frac{|\psi(w) - \psi(w_K^\star)|}{|w - w_K^\star|} \geq \frac{3}{4}|\psi'(w_K^\star)|. \tag{244}$$

Now choose any $w^\dagger$ such that

$$0 < |w^\dagger - w_K^\star| \leq r_2 \tag{245}$$

and such that

$$z_K^\dagger := (g_K^\star, w^\dagger, h_K^\star) \in D, \tag{246}$$

which is possible because $D$ is open and $(g_K^\star, w_K^\star, h_K^\star) \in D$. Let $Z_\star^\dagger$ denote the configuration obtained from $Z^\star$ by replacing the $K$th triple $(g_K^\star, w_K^\star, h_K^\star)$ with $z_K^\dagger$; denote by $\Delta\phi_\star^\dagger$ the corresponding update, and let

$$\delta w_\star := w^\dagger - w_K^\star. \tag{247}$$

Then $\Delta\phi_\star^\dagger = \Delta\phi(Z(w^\dagger))$ and $\Delta\phi^\star = \Delta\phi(Z(w_K^\star))$, so by (244) and (238),

$$\frac{\|\Delta\phi_\star^\dagger - \Delta\phi^\star\|_2}{|\delta w_\star|} \; \geq \; \frac{|\psi(w^\dagger) - \psi(w_K^\star)|}{|\delta w_\star|} \; \geq \; \frac{3}{4}|\psi'(w_K^\star)| \; \geq \; \frac{3}{4\sqrt{d_h}}\frac{\sqrt{\alpha}G_{\mathrm{gap}}}{(S^\star)^2}. \tag{248}$$

Define

$$K_4 := \frac{3}{4\sqrt{d_h}}. \tag{249}$$

We have thus found a *deterministic* pair of configurations $(Z^\star, Z_\star^\dagger)$ such that

$$\frac{\|\Delta\phi_\star^\dagger - \Delta\phi^\star\|_2}{|\delta w_\star|} \; \geq \; K_4\,\frac{\sqrt{\alpha}G_{\mathrm{gap}}}{(S^\star)^2}. \tag{250}$$

**Step 3: From a good point to a positive-probability neighborhood.** We now connect the deterministic pair $(Z^\star, Z_\star^\dagger)$ from Step 2 to the random resampled configuration of Definition C.7, and turn the pointwise lower bound (250) into a positive-probability event.

*Step 3(a). A basic continuity fact.* We first recall the simple fact from real analysis that we will use repeatedly:

*Let a function $F : \mathbb{R}^m \to \mathbb{R}$ be continuous, and suppose $F(x_0) > 0$ at some point $x_0 \in \mathbb{R}^m$. Then there exists a neighborhood $B$ around $x_0$ such that $F(x) \geq F(x_0)/2$ for all $x \in B$.*

Indeed, set $\varepsilon := F(x_0)/2 > 0$ and use continuity of $F$ at $x_0$ to obtain $\delta > 0$ such that $\|x - x_0\| < \delta$ implies $|F(x) - F(x_0)| < \varepsilon$. Then $\|x - x_0\| < \delta$ implies

$$F(x) \; \geq \; F(x_0) - |F(x) - F(x_0)| \; \geq \; F(x_0) - \varepsilon \; = \; \frac{F(x_0)}{2}. \tag{251}$$

We will apply this to a function $F$ defined on the product space of configurations and resampled triples.

*Step 3(b). Defining a continuous finite-difference functional.* For any configuration $Z = (g_1, w_1, h_1, \ldots, g_N, w_N, h_N)$ and any triple $(\widetilde{g}_K, \widetilde{w}_K, \widetilde{h}_K)$, let $Z^\dagger$ denote the configuration obtained by replacing the $K$th triple of $Z$ with $(\widetilde{g}_K, \widetilde{w}_K, \widetilde{h}_K)$. Define

$$\delta w(Z, \widetilde{w}_K) := \widetilde{w}_K - w_K, \qquad S(Z) := \sum_{i=1}^{N} w_i, \tag{252}$$

and consider the scalar functional

$$F(Z, \widetilde{g}_K, \widetilde{w}_K, \widetilde{h}_K) := \frac{S(Z)^2}{\sqrt{\alpha}G_{\mathrm{gap}}} \cdot \frac{\|\Delta\phi(Z^\dagger) - \Delta\phi(Z)\|_2}{|\delta w(Z, \widetilde{w}_K)|}, \tag{253}$$

defined on the domain where all weights are positive and $\delta w(Z, \widetilde{w}_K) \neq 0$. With the definition of $F$, we can see that (250) is

$$F(Z^\star, g_K^\star, w^\dagger, h_K^\star) \; \geq \; K_4 > 0. \tag{254}$$

We also need to show $F$ is continuous. By the explicit formula

$$\Delta\phi(Z) = \frac{1}{S(Z)}A(Z) - \frac{1}{S(Z)^2}B(Z), \tag{255}$$

with $A(Z)$ and $B(Z)$ linear in $(g_i, w_i, h_i)$, the map $Z \mapsto \Delta\phi(Z)$ is continuous whenever $S(Z) > 0$. The same is true for $Z^\dagger$. The maps $(Z, \widetilde{g}_K, \widetilde{w}_K, \widetilde{h}_K) \mapsto S(Z)$ and $(Z, \widetilde{g}_K, \widetilde{w}_K, \widetilde{h}_K) \mapsto \delta w(Z, \widetilde{w}_K)$ are also continuous. Thus $F$ is continuous at any point where $S(Z) > 0$, $\widetilde{w}_K > 0$, and $\delta w(Z, \widetilde{w}_K) \neq 0$.

At the point

$$(Z^\star, g_K^\star, w^\dagger, h_K^\star), \tag{256}$$

constructed in Step 2, we have $S(Z^\star) = S^\star > 0$, $\widetilde{w}_K = w^\dagger > 0$, and $\delta w(Z^\star, w^\dagger) = \delta w_\star := w^\dagger - w_K^\star \neq 0$, so $F$ is continuous there.

*Step 3(c). A product neighborhood where $F$ stays large.* We now apply the basic continuity fact from Step 3(a) to the function $F$ at the point

$$x_0 := (Z^\star, g_K^\star, w^\dagger, h_K^\star). \tag{257}$$

Since $F$ is continuous at $x_0$ and $F(x_0) \geq K_4$, there exists a neighborhood

$$N \subset \left( \left( \mathbb{R}^{d_g} \times (0, \infty) \times \mathbb{R}^{d_h} \right)^N \times \mathbb{R}^{d_g + 1 + d_h} \right) \tag{258}$$

of $x_0$ such that

$$F(Z, \widetilde{g}_K, \widetilde{w}_K, \widetilde{h}_K) \geq \frac{K_4}{2} \quad \text{for all } (Z, \widetilde{g}_K, \widetilde{w}_K, \widetilde{h}_K) \in N. \tag{259}$$

The product space $\left( \mathbb{R}^{d_g} \times (0, \infty) \times \mathbb{R}^{d_h} \right)^N \times \mathbb{R}^{d_g + 1 + d_h}$ can be factorized into open sets $U \times V$, where $U \subset \left( \mathbb{R}^{d_g} \times (0, \infty) \times \mathbb{R}^{d_h} \right)^N$, $V \subset \mathbb{R}^{d_g + 1 + d_h}$ are open. Recall that $F(\cdot) > K_4$ on $(Z^\star, g_K^\star, w^\dagger, h_K^\star)$. Therefore, we may choose two specific open sets $U$ and $V$ such that $(Z^\star, g_K^\star, w^\dagger, h_K^\star) \in U \times V$, and that for every $Z \in U$ and every $(\widetilde{g}_K, \widetilde{w}_K, \widetilde{h}_K) \in V$,

$$F(Z, \widetilde{g}_K, \widetilde{w}_K, \widetilde{h}_K) \geq \frac{K_4}{2}. \tag{260}$$

Finally, the map

$$(Z, \widetilde{w}_K) \mapsto \delta w(Z, \widetilde{w}_K) = \widetilde{w}_K - w_K \tag{261}$$

is continuous, and at $(Z^\star, w^\dagger)$ we have $\delta w_\star \neq 0$. Hence there exists a smaller product neighborhood $U' \times V'$ with $U' \subset U$ and $V' \subset V$, still containing $(Z^\star, g_K^\star, w^\dagger, h_K^\star)$, such that

$$\delta w(Z, \widetilde{w}_K) \neq 0 \quad \text{for all } Z \in U', \ (\widetilde{g}_K, \widetilde{w}_K, \widetilde{h}_K) \in V'. \tag{262}$$

On this smaller neighborhood, $F$ is well-defined and the bound (260) continues to hold. In particular, for all $Z \in U'$ and $(\widetilde{g}_K, \widetilde{w}_K, \widetilde{h}_K) \in V'$,

$$\frac{\|\Delta\phi(Z^\dagger) - \Delta\phi(Z)\|_2}{|\delta w(Z, \widetilde{w}_K)|} \geq \frac{K_4}{2} \frac{\sqrt{\alpha} G_{\text{gap}}}{S(Z)^2}. \tag{263}$$

*Step 3(d). Positive probability.* By Assumption C.8, the joint density of $Z$ is strictly positive on $D^N$ and $Z^\star \in D^N$, so

$$P(Z \in U') > 0. \tag{264}$$

Similarly, the resampled triple $(\widetilde{g}_K, \widetilde{w}_K, \widetilde{h}_K)$ has density $f > 0$ on $D$ and $(g_K^\star, w^\dagger, h_K^\star) \in D$, so

$$P\big((\widetilde{g}_K, \widetilde{w}_K, \widetilde{h}_K) \in V'\big) > 0. \tag{265}$$

Since $(\widetilde{g}_K, \widetilde{w}_K, \widetilde{h}_K)$ is independent of $Z$,

$$P\big(Z \in U', \ (\widetilde{g}_K, \widetilde{w}_K, \widetilde{h}_K) \in V'\big) = P(Z \in U') \, P\big((\widetilde{g}_K, \widetilde{w}_K, \widetilde{h}_K) \in V'\big) > 0. \tag{266}$$

On this event, the inequality (260) holds and $S(Z) > 0$, so by the definition of $F$ we have

$$\frac{\|\Delta\phi(Z^\dagger) - \Delta\phi(Z)\|_2}{|\delta w(Z, \widetilde{w}_K)|} \geq \frac{K_4}{2} \frac{\sqrt{\alpha} G_{\text{gap}}}{S(Z)^2}. \tag{267}$$

Recalling that in the resampling construction we write $\widetilde{\widetilde{\Delta\phi}} = \Delta\phi(Z^\dagger)$ and $\Delta\phi = \Delta\phi(Z)$, and setting $K_3 := K_4/2 = 3/(8\sqrt{d_h})$, we obtain

$$P\left(\frac{\|\widetilde{\widetilde{\Delta\phi}} - \Delta\phi\|_2}{|\delta w|} \geq K_3 \frac{\sqrt{\alpha}G_{\text{gap}}}{S^2}\right) \geq P\big(Z \in U', \ (\widetilde{g}_K, \widetilde{w}_K, \widetilde{h}_K) \in V'\big) > 0, \tag{268}$$

which is the desired positive-probability lower bound.

$\square$

# D. Theorem 3.6 on the Upper Bound of GSNR

## D.1. Common Notation

For the convenience of the reader, we restate some notations used in this section. These notations are consistent throughout the main text and the appendix of the paper.

Fix an integer $N \geq 1$. For each $i \in \{1, \ldots, N\}$ we are given random variables

$$w_i \in (0, \infty), \quad g_i \in \mathbb{R}^{d_g}, \quad h_i \in \mathbb{R}^{d_h}, \quad \gamma_i \in \mathbb{R}^{d_g}. \tag{269}$$

Define

$$\bar{\gamma} := \frac{1}{N}\sum_{i=1}^{N}\gamma_i, \qquad S := \sum_{i=1}^{N} w_i > 0, \qquad p_i := \frac{w_i}{S}, \qquad h_{\text{sum}} := \sum_{j=1}^{N} h_j. \tag{270}$$

As introduced in (10), the random gradient vector $\Delta\phi \in \mathbb{R}^{d_h}$ is given by

$$\Delta\phi = \frac{1}{S}\bar{\gamma}^\top \sum_{i=1}^{N} g_i\Big(h_i^\top - \frac{w_i}{S}\sum_{j=1}^{N} h_j^\top\Big) = \frac{1}{S}\bar{\gamma}^\top \sum_{i=1}^{N} g_i\big(h_i^\top - p_i h_{\text{sum}}^\top\big). \tag{10}$$

For convenience, define

$$A := \bar{\gamma}^\top \sum_{i=1}^{N} g_i h_i^\top \in \mathbb{R}^D, \qquad g_{\text{sum}} := \sum_{i=1}^{N} w_i g_i, \qquad B := \bar{\gamma}^\top g_{\text{sum}} h_{\text{sum}}^\top \in \mathbb{R}^D. \tag{271}$$

Then (10) can be rewritten as

$$\Delta\phi = \frac{1}{S}A - \frac{1}{S^2}B. \tag{272}$$

We measure the variance of $\Delta\phi$ by

$$\text{Var}(\Delta\phi) := \mathbb{E}\big[\|\Delta\phi - \mathbb{E}[\Delta\phi]\|_2^2\big] = \text{tr}\big(\text{Cov}(\Delta\phi)\big). \tag{273}$$

The gradient signal-to-noise ratio (GSNR) is

$$\text{GSNR}(\Delta\phi) := \frac{\|\mathbb{E}[\Delta\phi]\|_2^2}{\text{Var}(\Delta\phi)}. \tag{274}$$

## D.2. Common Assumptions

We work under the following additional assumptions.

**Assumption D.1** (Maximum range of random variables). The norms $\|\gamma_i\|_2, \|g_i\|_2, \|h_i\|_2$ and $w_i$ are bounded.

$$\text{For all } i, \ \|\gamma_i\|_2 + \|g_i\|_2 + \|h_i\|_2 + |w_i| < \infty \tag{275}$$

*Remark* D.2. Assumption D.1 may seem slightly strong in a mathematical sense, though it is always true in an engineering setting (since there is a maximum value for floating-point numbers represented by a finite number of bits). Strictly speaking, it is not needed. With assumptions on higher moments of the random variables, we can derive a bound similar to (302), which would include a higher order of $N$. We choose the simpler path that yields a tighter bound.

**Assumption D.3** (Negative moments of $S$).

$$\mathbb{E}[S^{-4}] < \infty. \tag{276}$$

**Assumption D.4** (Remaining variance of the dominant weight). Let $\mathcal{G} := \sigma\big((w_i, g_i, h_i)_{i \neq K}, (\gamma_i)_{i=1}^K\big)$ be the $\sigma$-field generated by all random variables excluding $w_K, g_K$ and $h_K$,

$$\mathrm{Var}(w_K \mid \mathcal{G}) = \sigma_w^2 > 0 \tag{277}$$

Given a fixed $s_\phi(\cdot)$ and a fixed $f_\theta(\cdot)$, $w_i$ and $h_i$ are both deterministic functions of $\phi$ and $x_i$, and $g_i$ is a deterministic function of $\theta$ and $x_i$. By Assumption C.1, $(x_i)$ are independent across $i$, so $w_i$ is independent from $w_j$, $h_j$, and $g_j$, for all $i \neq j$. Therefore, $\mathrm{Var}(w_K \mid \mathcal{G}) = \mathrm{Var}(w_K)$. Hence, this assumption boils down to $\mathrm{Var}(w_K) = \sigma_w^2 > 0$.

### D.3. A Lower Bound on $\mathrm{Var}(\Delta\phi)$

**Lemma D.5.** *Suppose Assumption C.1, and Assumptions D.1-D.4 hold, and the necessary assumptions for Corollary 3.5 hold. Then*

$$\mathrm{Var}(\Delta\phi) \geq C_{Lip}^2 G_{gap}^2 \sigma_w^2 \, \mathbb{E}\left[S^{-4} \mathbb{1}_{E_{Lip}}\right], \tag{278}$$

*where $\mathbb{1}_{E_{Lip}}$ is the indicator of the event $E_{Lip}$, on which the lower bound for $\frac{\|\Delta\phi - \widetilde{\widetilde{\Delta\phi}}\|_2}{|w_K - \widetilde{w}_K|}$ from Corollary 3.5 holds.*

*Proof.* The core idea of the proof is as follows: when the lower bound on $L_K = \frac{\|\Delta\phi - \widetilde{\widetilde{\Delta\phi}}\|_2}{|w_K - \widetilde{w}_K|}$ from Corollary 3.5 holds, we can related $\mathrm{Var}(w_K)$ with $\mathrm{Var}(\Delta\phi)$. As $w_k$ and $\widetilde{w}_K$ are i.i.d., $\mathbb{E}[(w_k - \widetilde{w}_K)^2] = 2\mathrm{Var}(w_K)$. Applying the same idea, we can lower bound $\mathrm{Var}(\Delta\phi)$. Here we need to condition on $\mathcal{G}$, $\mathbb{E}[\|\Delta\phi - \widetilde{\widetilde{\Delta\phi}}\|_2] = 2\mathrm{Var}(\Delta\phi \mid \mathcal{G})$.

Let $K$ be the dominant index, and $\mathcal{G} := \sigma\big((w_i, g_i, h_i)_{i \neq K}, (\gamma_i)_{i=1}^K\big)$ be the $\sigma$-field generated by all random variables excluding $w_K, g_K$ and $h_K$. By Assumption C.1, conditionally on $\mathcal{G}$ the only remaining randomness is in the $K$-th block $(w_K, g_K, h_K)$.

Given $\mathcal{G}$, we resample training data point $(x_K, y_K)$, which yields a new tuple $(\widetilde{w}_K, \widetilde{g}_K, \widetilde{h}_K)$, and the new vector $\widetilde{\widetilde{\Delta\phi}}$. By Corollary 3.5, on some event $E_{\mathrm{LIP}}$ with positive probability and a constant $C_{\mathrm{Lip}} > 0$, which is independent of $N$, we have

$$\frac{\|\Delta\phi - \widetilde{\widetilde{\Delta\phi}}\|_2}{|w_K - \widetilde{w}_K|} \geq \frac{C_{\mathrm{Lip}} G_{\mathrm{gap}}}{S^2}. \tag{279}$$

Squaring and taking conditional expectation given $\mathcal{G}$,

$$\mathbb{E}\left[\|\Delta\phi - \widetilde{\widetilde{\Delta\phi}}\|_2^2 \mid \mathcal{G}\right] \geq \frac{C_{\mathrm{Lip}}^2 G_{\mathrm{gap}}^2}{S^4} \, \mathbb{E}\left[(w_K - \widetilde{w}_K)^2 \mid \mathcal{G}\right] \mathbb{1}_{E_{\mathrm{LIP}}}, \tag{280}$$

where $\mathbb{1}_{E_{\mathrm{Lip}}}$ is the indicator function that the lower bound of (279) happens

$$\mathbb{1}_{E_{\mathrm{Lip}}} := \begin{cases} 1 & \text{if } E_{\mathrm{Lip}} \text{ happens.} \\ 0 & \text{otherwise.} \end{cases} \tag{281}$$

$w_K$ and $\widetilde{w}_K$ are i.i.d. (whether conditioned on $\mathcal{G}$ or not), so

$$\mathbb{E}[(w_K - \widetilde{w}_K)^2 \mid \mathcal{G}] = 2\,\mathrm{Var}(w_K \mid \mathcal{G}). \tag{282}$$

Conditioning on $\mathcal{G}$ makes $\Delta\phi$ and $\Delta\phi'$ i.i.d., so

$$\mathbb{E}\left[\|\Delta\phi - \widetilde{\widetilde{\Delta\phi}}\|_2^2 \mid \mathcal{G}\right] = 2\mathrm{Var}(\Delta\phi \mid \mathcal{G}) \tag{283}$$

$$\mathrm{Var}(\Delta\phi \mid \mathcal{G}) \geq \frac{c_{SP}^2}{4S^4} \, \mathrm{Var}(w_K \mid \mathcal{G}) \mathbb{1}_{E_{\mathrm{Lip}}}. \tag{284}$$

By Assumption D.4, $\mathrm{Var}(w_K \mid \mathcal{G}) = \mathrm{Var}(w_K) = \sigma_w^2 > 0$, so

$$\mathrm{Var}(\Delta\phi \mid \mathcal{G}) \ \geq \ \frac{C_{\mathrm{Lip}}^2 G_{\mathrm{gap}}^2 \sigma_w^2}{S^4} \, \mathbb{1}_{E_{\mathrm{Lip}}}. \tag{285}$$

Taking total expectation and using the law of total variance,

$$\mathrm{Var}(\Delta\phi) \ \geq \ \mathbb{E}\left[\mathrm{Var}(\Delta\phi \mid \mathcal{G})\right] \ \geq \ C_{\mathrm{Lip}}^2 G_{\mathrm{gap}}^2 \sigma_w^2 \, \mathbb{E}\left[S^{-4}\mathbb{1}_{E_{\mathrm{Lip}}}\right], \tag{286}$$

which is the desired bound. $\qquad\square$

### D.4. Upper Bound on $\|\,\mathbb{E}[\Delta\phi]\|_2^2$ and SNR

We now bound the numerator of the SNR.

**Lemma D.6.** *Under Assumptions C.1 and D.1-D.3, there exists a constant $C_m < \infty$ independent of $N$ such that*

$$\|\,\mathbb{E}[\Delta\phi]\|_2^2 \ \leq \ C_m \, N^4 \, \mathbb{E}[S^{-2}]. \tag{287}$$

*Proof.* From (10),

$$\Delta\phi = \frac{1}{S} \bar{\gamma}^\top \sum_{i=1}^{N} g_i(h_i^\top - p_i h_{\mathrm{sum}}^\top). \tag{288}$$

Taking norms and with triangle inequality,

$$\|\Delta\phi\|_2 \ \leq \ \frac{1}{S} \sum_{i=1}^{N} |\bar{\gamma}^\top g_i| \|h_i - p_i h_{\mathrm{sum}}\|_2. \tag{289}$$

Using Cauchy-Schwarz,

$$\|\Delta\phi\|_2 \ \leq \ \frac{1}{S} \|\bar{\gamma}\|_2 \sum_{i=1}^{N} \|g_i\|_2 \, \|h_i - p_i h_{\mathrm{sum}}\|_2. \tag{290}$$

Therefore

$$\|\Delta\phi\|_2^2 \ \leq \ \frac{1}{S^2} \|\bar{\gamma}\|_2^2 \Big(\sum_{i=1}^{N} \|g_i\|_2 \, \|h_i - p_i h_{\mathrm{sum}}\|_2\Big)^2. \tag{291}$$

Applying Cauchy-Schwarz to the sum,

$$\Big(\sum_{i=1}^{N} \|g_i\|_2 \, \|h_i - p_i h_{\mathrm{sum}}\|_2\Big)^2 \ \leq \ \Big(\sum_{i=1}^{N} \|g_i\|_2^2\Big)\Big(\sum_{i=1}^{N} \|h_i - p_i h_{\mathrm{sum}}\|_2^2\Big). \tag{292}$$

Hence

$$\|\Delta\phi\|_2^2 \ \leq \ \frac{1}{S^2} \|\bar{\gamma}\|_2^2 \Big(\sum_{i=1}^{N} \|g_i\|_2^2\Big)\Big(\sum_{i=1}^{N} \|h_i - p_i h_{\mathrm{sum}}\|_2^2\Big). \tag{293}$$

By Assumption D.1, we define finite constants $\Gamma_{\max}, G_{\max}, H_{\max} < \infty$ such that, for all $i$,

$$\|\gamma_i\|_2 \leq \Gamma_{\max}, \qquad \|g_i\|_2 \leq G_{\max}, \qquad \|h_i\|_2 \leq H_{\max} \tag{294}$$

First, by convexity of the norm,

$$\|\bar{\gamma}\|_2 = \Big\|\frac{1}{N} \sum_{i=1}^{N} \gamma_i\Big\|_2 \ \leq \ \frac{1}{N} \sum_{i=1}^{N} \|\gamma_i\|_2 \ \leq \ \Gamma_{\max}. \tag{295}$$

Next, $\|g_i\|_2 \leq G_{\max}$ for all $i$.

For the $h$-terms, we first bound

$$\|h_{\text{sum}}\|_2 = \Big\| \sum_{j=1}^{N} h_j \Big\|_2 \ \leq \ \sum_{j=1}^{N} \|h_j\|_2 \ \leq \ NH_{\max}. \tag{296}$$

Since $p_i = w_i/S \in (0,1]$, we have

$$\|h_i - p_i h_{\text{sum}}\|_2 \ \leq \ \|h_i\|_2 + |p_i| \, \|h_{\text{sum}}\|_2 \ \leq \ H_{\max} + NH_{\max} = (N+1)H_{\max}, \tag{297}$$

for all $i$.

Substituting these bounds into (293), we obtain

$$\|\Delta\phi\|_2 \ \leq \ \frac{1}{S} \Gamma_{\max} \sum_{i=1}^{N} G_{\max}(N+1)H_{\max} = \frac{1}{S} \Gamma_{\max} G_{\max} H_{\max} N(N+1). \tag{298}$$

Squaring both sides yields

$$\|\Delta\phi\|_2^2 \ \leq \ \frac{\Gamma_{\max}^2 G_{\max}^2 H_{\max}^2}{S^2} \, N^2 (N+1)^2. \tag{299}$$

For $N \geq 1$ we have $(N+1)^2 \leq 4N^2$, hence

$$\|\Delta\phi\|_2^2 \ \leq \ \frac{4\,\Gamma_{\max}^2 G_{\max}^2 H_{\max}^2}{S^2} \, N^4. \tag{300}$$

Taking expectations and using Assumption D.3,

$$\mathbb{E}\left[\|\Delta\phi\|_2^2\right] \ \leq \ 4\,\Gamma_{\max}^2 G_{\max}^2 H_{\max}^2 \, N^4 \, \mathbb{E}[S^{-2}]. \tag{301}$$

Finally, by Jensen's inequality for the convex function $x \mapsto \|x\|_2^2$,

$$\| \mathbb{E}[\Delta\phi]\|_2^2 \ \leq \ \mathbb{E}\left[\|\Delta\phi\|_2^2\right] \ \leq \ C_m N^4 \, \mathbb{E}[S^{-2}], \tag{302}$$

where $C_m(N) := 4\,\Gamma_{\max}^2 G_{\max}^2 H_{\max}^2 > 0$ is independent of $N$ and $w_i$. $\qquad\square$

### D.5. Theorem 3.6

Combining Lemmas D.5 and D.6 yields an SNR upper bound in terms of moments of $S$ and $G_{\text{gap}}^2$.

**Theorem 3.6** (SNR upper bound). *Under Assumptions C.1, D.1-D.4,*

$$\text{SNR}(\Delta\phi) := \frac{\| \mathbb{E}[\Delta\phi]\|_2^2}{\text{Var}(\Delta\phi)} \ \leq \ C_{\text{SNR}} \frac{N^4 \, \mathbb{E}[S^{-2}]}{G_{gap}^2 \, \mathbb{E}\left[S^{-4}\mathbb{1}_{E_{Lip}}\right]}, \tag{303}$$

*where $C_{\text{SNR}} = \frac{C_m}{C_{Lip}^2 \sigma_{w_i}} > 0$ is independent of $N$ and $G_{gap}^2$ .*

## E. Training Dynamics across Batch Sizes 512 and 1024

In Figure 4, we provide additional training dynamics for larger batch sizes, $N = 512$ and $N = 1024$, complementing Figure 2. The results exhibit trends consistent with those in Figure 2. First, the distribution of $p$ is quite spiky. Second, the empirical mean of $w_i$ decreases during training. Third, the sample variance of $p_i$, calculated as $\hat{\sigma}_p^2 := \frac{1}{N} \sum_{i=1}^{N}(p_i - 1/N)^2$, increases. Simultaneously, the effective batch size (EBS), calculated as $\left(\sum_i p_i^2\right)^{-1}$ decreases. These observations corroborate our theoretical analysis of data weights dynamics in Section 3.3.

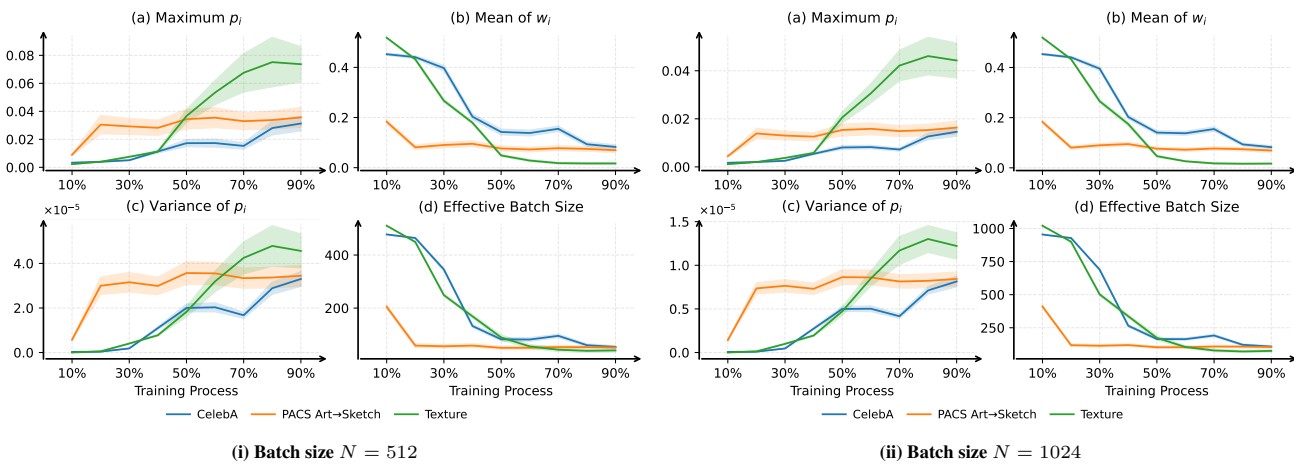

*Figure 4.* Training dynamics under batch sizes 512 and 1024: (a) the mean *unnormalized* weight $\hat{\mathbb{E}}[w_i] = S/N$, which decreases over time, (b) the maximum *normalized* weight $\max_i p_i$, which increases but to a lower ceiling for $N = 1024$ (c) the variance $\text{Var}(p_i)$, which increases to a lower ceiling for $N = 1024$, and (d) the effective batch size $B_{\text{eff}} = (\sum_i p_i^2)^{-1}$, which is larger for $N = 1024$. Colors indicate datasets. Shaded regions show mean $\pm$ std across independent batches.

# F. Details of the Datasets

We evaluate all methods on four benchmarks designed to assess robustness to different forms of spurious correlation and distributional variation. Dataset statistics refers to Table 5.

1. *Waterbirds* (Sagawa et al., 2020) evaluates robustness to spurious background correlations. The training set contains only land birds on forest backgrounds and waterbirds on water backgrounds, inducing a correlation between bird type and background. Validation and test sets are balanced across bird types and backgrounds.

2. *CelebA* (Yuan et al., 2024) evaluates spurious correlations between hair color and gender. We subsample the training set to contain only blonde females and non-blonde males. Validation and test sets are balanced across both attributes.

3. *Texture* (Yuan et al., 2024) evaluates texture bias using cue-conflict images from CCS (Geirhos et al., 2019), in which shape and texture cues disagree. The training set consists of standard ImageNet images restricted to the CCS classes. The CCS dataset is split into validation and test sets using a 25%/75% ratio for evaluation.

4. *PACS* (Li et al., 2017) evaluates domain generalization across four visual domains. We use the single-source single-target setting, resulting in 12 ordered source→target pairs. We train a separate model for each pair. For example, in the Art→Photo setting, training data is from the Art domain, while validation and test data are from the Photo domain. In Table 1, the four sub-columns (Art/Cartoon/Photo/Sketch) group these pairwise results by source domain: each entry is the average accuracy over the three target domains corresponding to that source. For example, the value under *Art* is the average performance across the settings Art→Photo, Art→Cartoon, and Art→Sketch.

# G. Details of the Baselines

We provide implementation details for all the baselines considered in this paper. In hard selection, we discard a fixed fraction $p\%$ of training samples with the lowest scores, corresponding to those deemed less informative. We evaluate $p \in \{5, 10, 20, 30\}$ for each dataset and consistently observe that $p = 5\%$ yields the best performance. We therefore adopt this setting throughout. In soft selection, all training samples are retained and reweighted according to their scores.

**Training-Dynamics Heuristics.** We first train a classifier on the full training set without data selection and record its training dynamics. We consider two per-sample signals: *forgetting events* and *gradient norms*. For each sample $i$ in soft selection, we convert the score $s_i$ into a weight in $(0, 1]$ as $w_i = 1/(1 + \log(1 + s_i))$, so that samples with higher forgetting counts receive smaller weights. This monotonic transformation preserves the ranking induced by the original scores while keeping the resulting weights bounded.

| Dataset | Train | | Validation | Test |
|---|---|---|---|---|
| | **Real** | **Synthetic** | | |
| Waterbirds | 1139 | 839 | 1439 | 5794 |
| CelebA-sub | 3535 | 3535 | 500 | 1000 |
| Texture | 9600 | 9600 | 320 | 960 |
| PACS Art Painting | 1840 | 1840 | 208 | 2048 |
| PACS Cartoon | 2107 | 2107 | 237 | 2344 |
| PACS Photo | 1499 | 1499 | 171 | 1670 |
| PACS Sketch | 3531 | 3531 | 398 | 3929 |

*Table 5.* Dataset statistics for training, validation, and test splits.

*Forgetting Events.* For each sample, we compute a forgetting count, defined as the number of times the model's prediction for that sample transitions from correct to incorrect during training, following Toneva et al. (2019).

*GraNd (Gradient Norm).* For each sample, we compute its GraNd score as the $\ell_2$ norm of the gradient of the sample loss with respect to the model parameters, following Paul et al. (2021).

**Synthetic Image Selection Heuristics.** These heuristics score synthetic samples according to their semantic alignment with the target label or their predictive confidence under an auxiliary model.

*CLIP Similarity.* For each sample $i$, we compute the cosine similarity between its CLIP image embedding and the CLIP text embedding of its ground-truth label. We denote this score by $s_i^{\mathrm{CLIP}}$ and map it to a non-negative weight as $w_i = (1 + s_i^{\mathrm{CLIP}})/2$. This linear transformation preserves the score ranking used for hard selection and ensures non-negative weights for stable soft reweighting.

*Auxiliary Classifier Confidence (Aux-Clf).* We train an auxiliary classifier $f_{\mathrm{aux}}$ on the real training set and use its predicted probability for the ground-truth label as the sample weight: $w_i = p_{f_{\mathrm{aux}}}(y_i \mid x_i)$. For *Aux-Clf (Val)*, we instead train the auxiliary classifier on the labeled validation set, which provides a stronger signal of alignment with the target distribution.

*CLIP Similarity.* For each sample $i$, we compute the cosine similarity between its CLIP image embedding and the CLIP text embedding of its ground-truth label. We map this similarity to $[0, 1]$ using $w_i = \frac{1 + s_i^{\mathrm{CLIP}}}{2}$. This transformation is linear and strictly monotonic. It preserves the ranking of samples for hard selection and avoids negative weights during soft reweighting.

**Domain Adaptation Methods.** These methods assess how closely each training sample matches the target (validation) distribution.

*NormSim.* We compute CLIP image embeddings for all training samples and validation samples. For each training sample $x_i$, we define its weight as the $\ell_2$ norm of its similarity vector to all validation samples:

$$w_i = \mathrm{NormSim}(x_i) = \left\| \left[ \langle f(x_i), f(x_1^{\mathrm{val}}) \rangle, \ldots, \left\langle f(x_i), f(x_{|\mathcal{D}_{\mathrm{val}}|}^{\mathrm{val}}) \right\rangle \right] \right\|_2,$$

where $|\mathcal{D}_{\mathrm{val}}|$ denotes the number of validation samples. This score measures how strongly $x_i$ aligns with the target validation distribution in CLIP space.

*Importance Weighting.* We follow Choi et al. (2021), using CLIP embeddings in place of the learned features used in their original implementation. Specifically, we train a binary classifier $c : \mathcal{X} \to [0, 1]$ to distinguish training samples from validation samples, where training samples are labeled as $y = 0$ and validation samples as $y = 1$. For each training sample $x_i$, we estimate its importance weight by the density ratio between the validation and training distributions:

$$w_i = r(x_i) = \frac{p(x_i \mid y_i = 1)}{p(x_i \mid y_i = 0)} = \frac{c^*(x_i)}{1 - c^*(x_i)},$$

where $c^*(x_i)$ denotes the probability assigned by the optimal classifier that $x_i$ comes from the validation distribution.

| Parameter | Value |
|-----------|-------|
| Inference steps | 30 |
| Image strength | 0.75 |
| Guidance scale | 2.0 |
| Sampler | UniPC |

*Table 6.* Generator hyperparameters used for image-conditioned editing with Stable Diffusion.

## H. Additional Experimental Details

The classifier is a pretrained ResNet-50 optimized using SGD with momentum and a learning rate of $1 \times 10^{-3}$. The selection network is a three-layer MLP with a hidden dimension of 100 and a sigmoid output, optimized using AdamW. During training, the two networks are updated in an alternating manner: each optimization step of the selection network is followed by one optimization step of the classifier. Unless otherwise specified, we use a batch size of $N = 1024$ and set $K = 3$ for the $K$ nearest neighbors used in input feature computation. We generate data following the original setup of Dunlap et al. (2023); the corresponding hyperparameters are provided in Table 6. We train the classifier following the original recipe of Yuan et al. (2024).

The generated data for WaterBirds are publicly available in Dunlap et al. (2023), and we directly use the released dataset. We generate the remaining three datasets, CelebA, Texture, and PACS, ourselves, as they are not publicly available. For prompt design on these datasets, we follow Yuan et al. (2024) and use dataset-specific textual prompts for image-conditioned editing with Stable Diffusion v1.5[4]:

*PACS.* Prompts follow the template "A [DOMAIN] of [CLASS LABEL]", where *DOMAIN* corresponds to the target PACS domain (Art Painting, Cartoon, Photo, or Sketch).

*CelebA.* Prompts are constructed to preserve the target label (hair color) while flipping the spurious attribute (gender). Specifically, for images labeled as *blonde*, we use the prompt "blonde male", and for images labeled as *non-blonde*, we use the prompt "non-blonde female".

*Texture.* Prompts follow the template "[STYLE] [CLASS LABEL]". We use the following set of style templates:

pointillism, rubin statue, rusty statue, ceramic, vaporwave, stained glass, wood statue, metal statue, bronze statue, iron statue, marble statue, stone statue, mosaic, furry, corel draw, simple sketch, stroke drawing, black ink painting, silhouette painting, black pen sketch, quickdraw sketch, grainy, surreal art, oil painting, fresco, naturalistic painting, stylised painting, watercolor painting, impressionist painting, cubist painting, expressionist painting, artistic painting

## I. Details of the Input Features

This section provides the details of the input features used by the selection network. For a training sample $(x_i, y_i)$, we use two types of representations. Let $h_i^{\mathrm{cls}}$ denote the online embedding extracted from the penultimate layer of the current classifier, and let $h_i^{\mathrm{gen}}$ denote the offline embedding extracted from the image generation model.

**Local Features.** Let $\mathcal{B}_{\mathrm{train}}$ and $\mathcal{B}_{\mathrm{val}}$ denote the current training batch and validation batch, respectively. Using online encodings $h^{\mathrm{cls}}$, we compute cosine similarities between $x_i$ and its $K$ nearest neighbors within the current batches:

$$\mathcal{S}_{\mathrm{train\text{-}batch}}(x_i) = \left\{ \cos\!\left(h_i^{\mathrm{cls}}, h_j^{\mathrm{cls}}\right) \mid j \in \mathcal{N}_K(x_i; \mathcal{B}_{\mathrm{train}}) \right\},$$

$$\mathcal{S}_{\mathrm{val\text{-}batch}}(x_i) = \left\{ \cos\!\left(h_i^{\mathrm{cls}}, h_j^{\mathrm{cls}}\right) \mid j \in \mathcal{N}_K(x_i; \mathcal{B}_{\mathrm{val}}) \right\},$$

where $\mathcal{N}_K(x_i; \mathcal{B})$ denotes the indices of the $K$ nearest neighbors of $x_i$ within set $\mathcal{B}$ under cosine distance.

Let $\mathcal{D}_{\mathrm{train}}$ and $\mathcal{D}_{\mathrm{val}}$ denote the whole training dataset and validation dataset, respectively. We use offline encodings $h^{\mathrm{gen}}$ to

---

[4]https://huggingface.co/stable-diffusion-v1-5/stable-diffusion-v1-5

compute dataset-level neighborhood features. We compute:

$$\mathcal{S}_{\text{train}}(x_i) = \left\{ \cos\!\left(h_i^{\text{gen}}, h_j^{\text{gen}}\right) \mid j \in \mathcal{N}_K(x_i; \mathcal{D}_{\text{train}}) \right\},$$

$$\mathcal{S}_{\text{val}}(x_i) = \left\{ \cos\!\left(h_i^{\text{gen}}, h_j^{\text{gen}}\right) \mid j \in \mathcal{N}_K(x_i; \mathcal{D}_{\text{val}}) \right\},$$

where $\mathcal{N}_K(x_i; \mathcal{D})$ denotes the indices of the $K$ nearest neighbors of $x_i$ within the whole dataset $\mathcal{D}$ under cosine distance.

We additionally compute the label agreement score (Zhang et al., 2024). Specifically, we first find the $K$ nearest neighbors of $x_i$ in the training set $\mathcal{D}_{\text{train}}$ using the offline embedding $h^{\text{gen}}$. The label agreement is then defined as the fraction of these neighbors that share the same label as $x_i$:

$$s_{\text{LA}}(x_i) = \frac{1}{K} \sum_{j \in \mathcal{N}_K(x_i; \mathcal{D}_{\text{train}})} \mathbb{1}[y_j = y_i].$$

**Global Features.** Let $\mathcal{D}_{\text{train}}^c$ and $\mathcal{D}_{\text{val}}^c$ denote the training set and validation set of class $c$, respectively. For each class $c$, we compute class centroids separately for the training and validation sets with offline embeddings:

$$\mu_{\text{train}}^c = \frac{1}{|\mathcal{D}_{\text{train}}^c|} \sum_{x_j \in \mathcal{D}_{\text{train}}^c} h_j^{gen}, \qquad \mu_{\text{val}}^c = \frac{1}{|\mathcal{D}_{\text{val}}^c|} \sum_{x_j \in \mathcal{D}_{\text{val}}^c} h_j^{gen}.$$

For each sample $x_i$ with label $y_i$, we compute its Euclidean distance and cosine similarity to both centroids:

$$s_{\ell_2}^{\text{train}}(x_i) = \|h_i^{gen} - \mu_{\text{train}}^{y_i}\|_2, \quad s_{\ell_2}^{\text{val}}(x_i) = \|h_i^{gen} - \mu_{\text{val}}^{y_i}\|_2,$$

$$s_{cos}^{\text{train}}(x_i) = \cos(h_i^{gen}, \mu_{\text{train}}^{y_i}), \quad s_{cos}^{\text{val}}(x_i) = \cos(h_i^{gen}, \mu_{\text{val}}^{y_i}).$$

In addition, we compute an online validation-batch centroid to capture the current target distribution during training. Let $\mathcal{B}_{\text{val}}$ denote the validation batch samples. Using online encodings $h^{\text{cls}}$, we compute

$$\mu_{\text{val-batch}} = \frac{1}{|\mathcal{B}_{\text{val}}|} \sum_{x_j \in \mathcal{B}_{\text{val}}} h_j^{\text{cls}}.$$

For each training sample $x_i$, we further compute its cosine similarity to the corresponding online validation-batch centroid:

$$s_{\cos}^{\text{val-batch}}(x_i) = \cos\!\left(h_i^{\text{cls}}, \mu_{\text{val-batch}}\right).$$

**Optimization Dynamics Features.** We train an auxiliary classifier without data selection and extract optimization-dynamics features, including *forgetting events* (Toneva et al., 2019), as well as the *gradient norm* and *error vector norm* (Paul et al., 2021), following the exactly original implementations in the respective papers.

**Class-Indicator Features.** We use a 4-dimensional learnable embedding for the class label $y_i$ and a separate 1-dimensional learnable embedding to indicate the data source (real vs. synthetic).

## J. Algorithm Details

In this section, we describe the implementation details of our method summarized in Algorithm 1. Following Liu et al. (2018), we reduce the computational cost of hypergradient computation by adopting a finite-difference approximation[5]. When training with large batch sizes across multiple GPUs, we compute the normalization term $S = \sum_i w_i$ using `torch.distributed.nn.functional.all_reduce`[6]. This operation has an autograd kernel registered and is therefore fully differentiable, allowing gradients to be correctly propagated across GPUs during backpropagation.

---

[5] https://en.wikipedia.org/wiki/Finite_difference
[6] https://github.com/pytorch/pytorch/blob/main/torch/distributed/nn/functional.py#L205

---

**Algorithm 1** One-step Meta-learning for Training-data Selection (MTS)

---

**Require:** Training set $\mathcal{D}_{\text{train}}$, validation set $\mathcal{D}_{\text{val}}$; learning rates $\eta_\theta, \eta_\phi$; finite-difference step $\epsilon > 0$; total iterations $T$

1: Initialize classifier parameters $\theta_0$ and selection network parameters $\phi_0$
2: **for** iteration $t = 0, 1, \ldots, T - 1$ **do**
3:     Sample a training batch $\mathcal{B}_{\text{train}} = \{(x_i, y_i)\}_{i=1}^N \subset \mathcal{D}_{\text{train}}$ and a validation batch $\mathcal{B}_{\text{val}} = \{(x_i^{\text{v}}, y_i^{\text{v}})\}_{i=1}^N \subset \mathcal{D}_{\text{val}}$
4:     Compute data weights $w_i = s_{\phi_t}(x_i)$ and normalization term $S = \sum_{i=1}^N w_i$
5:     Perform one-step look-ahead classifier update:

$$\theta_t' \leftarrow \theta_t - \eta_\theta \nabla_\theta \mathcal{L}_{\text{train}}(\theta_t, \phi_t), \quad \text{where} \quad \mathcal{L}_{\text{train}}(\theta_t, \phi_t) = \frac{1}{S} \sum_{i=1}^N w_i \, \ell(x_i, y_i, \theta_t)$$

6:     Compute validation gradient at the look-ahead point:

$$\nabla_\theta \mathcal{L}_{\text{val}}(\theta_t') = \frac{1}{N} \sum_{i=1}^N \nabla_\theta \ell(x_i^{\text{v}}, y_i^{\text{v}}, \theta_t')$$

7:     Perturb classifier parameters:

$$\theta^+ \leftarrow \theta_t + \epsilon \, \nabla_\theta \mathcal{L}_{\text{val}}(\theta_t'), \qquad \theta^- \leftarrow \theta_t - \epsilon \, \nabla_\theta \mathcal{L}_{\text{val}}(\theta_t')$$

8:     Compute training losses $\mathcal{L}_{\text{train}}(\theta^+, \phi_t)$ and $\mathcal{L}_{\text{train}}(\theta^-, \phi_t)$ on $\mathcal{B}_{\text{train}}$
9:     Estimate hypergradient via finite differences:

$$\nabla_{\phi_t} \leftarrow -\eta_\theta \frac{\nabla_\phi \mathcal{L}_{\text{train}}(\theta^+, \phi_t) - \nabla_\phi \mathcal{L}_{\text{train}}(\theta^-, \phi_t)}{2\epsilon}$$

10:     Update selection network:

$$\phi_{t+1} \leftarrow \phi_t - \eta_\phi \, \nabla_{\phi_t}$$

11:     Update classifier:

$$\theta_{t+1} \leftarrow \theta_t - \eta_\theta \nabla_\theta \mathcal{L}_{\text{train}}(\theta_t, \phi_{t+1})$$

12: **end for**
13: **return** classifier $f_{\theta_T}$

---

