# OpenReview forum: "On the Difficulty of Learning a Meta-network for Training Data Selection"
_ICML.cc/2026/Conference — ICML 2026 spotlight_

### Official Review · Reviewer_gPdC · 2026-02-27

**Soundness:** 3
**Presentation:** 2
**Significance:** 3
**Originality:** 3
**Overall Recommendation:** 4
**Confidence:** 3

**Summary:**

This work aims to mitigate the distributional match. With extensive theoretical proof and experiments on four datasets (WaterBirds, CelebA, Texture and PACS), the proposed method shows a large gain (+5.49%) over prior work, reaching at 73.75% average accuracy. Ablation studies demonstrate the contribution of the larger batch size practice and local+global features.

**Compliance With Llm Reviewing Policy:**

Affirmed.

**Final Justification:**

This work has conducted extensive experiments to demonstrate its effectiveness in high quality. The authors provided further clarity and resolved most of my concerns. I have raised the rating to __4: Weak accept__.

**Key Questions For Authors:**

It would be great to answer at least __SN_W2__ and __SN_W4__ (__P_W3__),

**Limitations:**

No.

Increasing batch size will directly increase GPU memory usage, resulting in more energy consumption and CO2 emissions. It would be great to discuss the tradeoff.

**Strengths And Weaknesses:**

__Soundness (SN)__

__Strengths__:

This work has conducted extensive experiments to justify their claims:

__SN_S1__: Figure 1: signal-to-noise ratio (GSNR) study.

__SN_S2__: Table 1: the proposed method appears a large performance gain over the second best method (+5.49%) across four datasets.

__SN_S3__: Figure 2: Training dynamics across two batch sizes.

__SN_S4__: Table 2: Ablation Studies demonstrate that each of the large batch size and the proposed Local and Global Feature encoding contributes to the performance gain.

__SN_S5__: This work provides extensive theoretical analysis to prove the importance of high GSNR and the relation to large batch size, therefore proposing increasing large batch size as one practical solution.

__Weaknesses__:

__SN_W1__: The method is tested on ResNet-50 but it might yield more insights and see if it can generalize to other visual architectures such as ViT (or Mamba but not required at the moment), as ConvNet (e.g. ResNet-50) focuses more on local connectivity while ViT favors global context.

__SN_W2__: __Local vs Global Features__. It would be beneficial to conduct ablation studies on the Local and Global Features proposed in this work to see how each contribute to the performance gain differently. This might potentially yield more insights in addition to __SN_W1__.

__SN_W3__: Following __SN_S3__, since only two batch sizes were evaluated, it would be beneficial to include at least three batch sizes to improve statistical robustness.

__SN_W4__: The scope of demonstrating the ability of generalization remain unclear, e.g. within domain (train/val/test gap in the same dataset) or cross domain (train/val and test sets in different datasets). This part is also detailed in __P_W3__.

---

__Presentation (P)__

__Strengths__:

__P_S1__: Overall, this work is presented with clear logic flow with high quality figures and tables.

__Weakness__:

__P_W1__ (Minor): It might be a good idea to keep the order consistent from Line 195 to 212 (Right) when describing Figure 2 and the order of subfigures. Specifically, the correspondences appear to be as follows: p -- (b), w -- (a), variance of p -- (c), effective batch size -- (d), thus the text from Line 195 to 212 describes Figure 2 in this order: (b)(a)(c)(d). As a result, a reader might need to switch between subfigures to follow the logical flow.

__P_W2__ (Minor): It is good to include ablation studies in Table 2. It might enhance the clarity to include the settings for the proposed method somewhere in the (Ours) row so that the changes are clear, such as adding batch size of N = 1024? Or the local and global features.

__P_W3__: The scope of “distributional mismatch” is unclear. The synthetic and real data example used at the beginning and the results in Figure 1 PACS Photo->Art seem to suggest cross-dataset distribution shift. But the results in Table 1 and the text (detailed as follows) seem to suggest only train/val distributional mismatch. It would be great to provide further clarity.

__P_W3.1__: Line 320 “For each benchmark, the training split differs from the validation and test splits, which share the same target distribution.” It would be great to provide further clarity what does “same target distribution” mean in the context of four datasets in Table 1?

__P_W3.2__: In Table 1, is each number computed within the same dataset? Are training and val/test sets in the same dataset or is there any cross over between datasets like D_train in WaterBirds but D_val|test in CelebA?

__P_W3.3__: This work compares against some domain adaptation methods (Line 358). In general domain adaptation methods are used to tackle the domain gap, e.g. aligning training distribution with testing distribution typically in different datasets (such as WaterBirds -> CeleA). But Line 2788 describes as “each training sample matches the target (validation)”, which seems to be within the same dataset. Does it specifically refer to only train/val adaptation instead of cross-dataset adaptation?

__P_W4__: (Minor) Table 1 and 2 appear to show accuracy but there’s no explicit mention regarding “accuracy” in these two tables.

---

__Significance (SF)__

__Strengths__:

__SF_S1__: It is great that this work distills two simple messages that a practitioners can use during training: (1) increasing batch size and (2) informative features (local + global).

__Weaknesses__:

__SF_W1__: In practice, increasing batch size could be constrained by the GPU memory limits. The default batch size in this work is 1024, which is extremely large in practical trainings. In my experience, a 128 or 256 batch size from ImageNet can easily dominate a 48GB GPU of A6000 when training a single network.

---

__Originality (OR)__

__Strengths__:

__OR_S1__: This work details how it differs from prior work when introducing its component and theoretical analysis throughout the paper.

---

> ### Author Rebuttal · Authors · 2026-03-30
>
> We thank reviewer gPdC for the diligence and thoughtful feedback.
>
> ---
>
> **SN\_W1​: Generalization to ViT.**
>
> We conduct experiments using ViT-Base-Patch32 pretrained on ImageNet-21K as the visual backbone. [[Results]](https://github.com/AnonymousPaperNo9458/Rebuttal/tree/main/ViTResults) show our method consistently outperforms the No Selection baseline and MetaWeightNet (MW-Net) by +5.30% and +5.76% on average, respectively, demonstrating that its effectiveness is not tied to a specific architecture.
>
> ---
>
> **SN\_W2​: Ablation on local and global features.**
>
> We conduct an ablation study, with [[results]](https://github.com/AnonymousPaperNo9458/Rebuttal/blob/main/AblationFeatures/readme.md). Removing either local or global features consistently degrades performance across datasets, with drops of 1.16% and 1.05%, respectively, demonstrating that both contribute to the effectiveness of our method.
>
> ---
>
> **SN\_W3​: Include at least three batch sizes in Figure 2.**
>
> In addition to $N$ = 64 and 256 shown in Figure 2, we supplement results for $N$ = 512 and $N$ = 1024 ([LINK](https://github.com/AnonymousPaperNo9458/Rebuttal/blob/main/TrainingDynamics/readme.md)). The figures exhibit the same trends predicted by our theoretical analysis: effective batch size decreasing, and max $p_i$ and variance of $p_i$ increasing over time. Further, a larger batch reduces these effects.
>
> ---
>
> **SN\_W4​, P\_W3.1, P\_W3.2, P\_W3.3: Clarification of experiment setting.**
>
> We thank the reviewer for pointing out this ambiguity and provide the following clarification.
>
> For Waterbirds, CelebA, and Texture, we train and evaluate within each dataset. There is no cross-dataset adaptation. The validation and test data share the same distribution (referred to as the target domain), while the training data comes from a different distribution (source domain). For example, in WaterBirds, the training set contains only land birds on forest backgrounds and waterbirds on water backgrounds, while validation and test sets are balanced across bird types and backgrounds.
>
> For PACS, we use the *single-source single-target* setting, resulting in 12 ordered source→target pairs. We train a separate model for each pair. For example, in the Art→Photo setting, training data is from the Art domain, while validation and test data are from the Photo domain. In Table 1, the four sub-columns (Art/Cartoon/Photo/Sketch) group these pairwise results by source domain: each entry is the average accuracy over the three target domains corresponding to that source. For example, the value under “Art” is the average performance across the settings Art→Photo, Art→Cartoon, and Art→Sketch.
>
> ---
>
> **P\_W1, P\_W2, P\_W4**
>
> We thank the reviewer and will: (1) fix Figure 2 ordering, (2) clarify numbers in Table 1/2 as accuracy, and (3) improve Table 2 annotations by specifying *Ours (N=1024, all features)*.
>
> ---
>
> **P\_W3: Clarification of “distributional mismatch”.**
>
> Our use of “distributional mismatch” refers to settings where the training distribution differs from the validation/test distribution. Such mismatch can arise from multiple sources. First, when synthetic data is used to augment training, the resulting training distribution may deviate from the real data distribution at test time. Second, in many real-world datasets, inherent properties such as spurious correlations and domain shifts also introduce discrepancies between training and test distributions. These scenarios collectively motivate the need for selective training under distributional mismatch.
>
> ---
>
> **SF\_W1​: Batche sizes are limited by GPU RAM.**
>
> We fully agree that large batches may be practically inconvenient, and this is precisely why the message of this paper is important. Out of convenience, many academic researchers may choose small batch sizes when training MTS. When it performs poorly under small batch sizes, they may conclude that MTS simply doesn’t work and abandon the research direction. By drawing attention to a potential blind spot, our paper may inspire further research on MTS.
>
> It is also worth noting that larger batch sizes do not necessitate greater memory footprints. For example, we can perform gradient accumulation (running multiple small batches before optimizer.step()), gradient checkpointing, and mixed-precision training. This paper may also motivate further research to reduce memory usage.
>
> Finally, even at moderate batch sizes like 256, our method achieves improvements of +4.42% and +3.84% over No Selection and MW-Net, respectively, while requiring only a single GPU.
>
> ---
>
> **Limitations: CO₂ emission and performance tradeoff.**
>
> We thank the reviewer for pointing this out and will add a limitation section to discuss the tradeoff. We visualize the relation between memory and accuracy [[here.]](https://github.com/AnonymousPaperNo9458/Rebuttal/blob/main/Computation-Performace/readme.md).

---

> > ### Author Rebuttal · Reviewer_gPdC · 2026-04-01
> >
> > Thank the authors for the detailed response, which has addressed most of the questions. I have raised my rating.

---

> > > ### Author Response · Authors · 2026-04-03
> > >
> > > Thank you very much for your thoughtful feedback and constructive suggestions. We are delighted that our rebuttal has addressed your concerns and we will incorporate the new results and discussion into the revised version. Thanks again for your time and attention.

---

### Official Review · Reviewer_Y8yx · 2026-03-10

**Soundness:** 3
**Presentation:** 4
**Significance:** 3
**Originality:** 3
**Overall Recommendation:** 5
**Confidence:** 4

**Summary:**

This paper deals with the meta-learning problem of estimating sample weights for training sample selection to train effective models (in this case, classifiers). The authors focus on a particular problem formulation, where weight selection is treated as bilevel optimization problem which is denoted as Meta-learning for Training Selection (MTS). The main motivation for this work is the observation that MTS leads to poor results, in some cases worse than the non-weighted baseline (i.e., without sample selection). The paper follows a formal approach which explores the behaviour of the Gradient Signal to Noise Ratio (GSNR) during MTS and identifies key issues, such as the dominance of certain samples during the weight selection procedure. The authors then identify that the GSNR is upper bounded by a quality that depends on the batch size, N. Increasing batch size therefore is proposed as an effective strategy towards improving the effectiveness of MTS. In addition to these results, the authors indicate that data quality plays an important role, and therefore propose a set of features for use by the weight selection network. Experiments indicate a diverse set of settings (e.g. spurious correlations, domain generalization), where real data are augmented by synthetic data.

**Compliance With Llm Reviewing Policy:**

Affirmed.

**Final Justification:**

In their rebuttal the authors provided additional data / evidence to support their claims, to the degree possible. Given these clarifications, I have updated my score.

**Key Questions For Authors:**

Please address (wherever possible) the comments in the "Weaknesses" section above, especially with respect to the link between the two parts of the submission and the empirical assessment of the method. These will help better assess the contribution and value of the proposed approach.

The experimental setup mentions pairwise comparisons for PACS, but then later (and in the experimental results section) it seems that the SDG setting is used (i.e., the average across the non-training domains is reported). This should be clarified.

**Limitations:**

A separate "limitations" section (could be part of the appendix) indicating known weak points and/or strong assumptions of the presented analysis would be helpful to the readers.

**Strengths And Weaknesses:**

### Strengths

 - Mathematical rigor: A clear strong point of the paper is that it approaches the problem formally, and provides a formal link between GSNR and batch size. Importantly, such formal derivations have the potential to drive future work in this area.

 - Well-motivated experimental design: The experimental setup uses a diverse set of datasets and competing methods, and synthetic data augmenting the real data to improve robustness.

 - Impoved performance of the proposed feature set against the baseline for large batch sizes.

 - Well-written and clearly structured paper: Despite the extensive formal arguments, most of the paper flows very well and is easy to understand.

### Weaknesses

 - The paper seems to be the concatenation of two distinct parts. The first part includes all the theoretical contribution and link between GSNR and batch size. The second part includes the proposed feature representation and its evaluation. These two contributions are only weakly linked. Importantly, although the first part has a strong theoretical grounding, the second part is more arbitrary / heuristic. An effort should be made by the authors to better reconcile these two parts.

 - The formalization using the damped ODE of Eqs. (15) and (16) looks like an assumption and also uses fixed $a_i$ (which MTS does not). These may be far from the actual training dynamics (as the authors somewhat admit), so this raises concerns about the usefulness of this analysis.

 - It is not very clear why assumption C.11 should hold. Perhaps some additional arguments or examples would be helpful here.

 - In the comparison with MetaWeightNet it is not evident if the improvements are due to the new feature set or the increased batch size. One would expect an additional experiment with N=1024 on MetaWeightNet.

 - In the ablation study the paper's arguments seem to be mainly supported by PACS and not the rest of the datasets. Taking the average across datasets to reach a conclusion is a well-known fallacy in the analysis of machine learning algorithms (see e.g., J. Demsar, JMLR 2006).

---

> ### Author Rebuttal · Authors · 2026-03-31
>
> We thank reviewer Y8yx for the diligence and thoughtful feedback.
>
> **W1: Connection between GSNR and features.**
>
> For MTS to work, the selection network must learn an effective function $s_\phi(x)$ that distinguishes good training data from poor training data. Two factors are at play here: the range of functions (the hypothesis space) that $s_\phi(x)$ can represent, and our ability to find the best function among possible functions.
>
> Increasing GSNR facilitates the search for an optimal function among the hypothesis space.  When GSNR is low, optimization becomes slow and unstable, which makes it difficult to find the optimal $s_\phi(x)$ in the hypothesis space.
>
> However, if the hypothesis space does not contain effective functions, even if we find the best function in that space, it is of no use. Extending the range of features expands the hypothesis space. Intuitively, the new features we propose correlate with the quality of training data, so the hypothesis space should contain some useful functions.
>
> As an analogy, we search for a good $s_\phi(x)$ like a needle in a haystack. Good GSNR makes sure our search is efficient, that we can check many blades quickly. Good features ensure that the haystack is large enough so that it probably contains a needle. We must have these two factors simultaneously. Otherwise the search will fail.
>
> In the real world, things are slightly more complex because GSNR is also related to generalization. But the above intuition still largely holds.
>
> ---
>
> **W2: ODE may not describe training accurately.**
>
> As George Box famously noted, *"All models are wrong, but some are useful"*. For theoretical analysis, we must balance between the complexity of the model and the usefulness of its predictions. Make the model too simple, and it will be too far from reality. Make it too complex, and the mathematical analysis becomes intractable.
>
> We choose to build a math model with $a_i$ fixed. If we want to model how $a_i$ changes, we will have to model how validation and training gradients change, which probably mandates fairly strong assumptions about the loss functions, and we would rather not do that.
>
> Importantly, the ODE with $a_i$ fixed allows us to make qualitatively useful predictions that agree with reality (Figure 2). As part of the rebuttal, we expand Figure 2 to include N=512 and N=1024 ([LINK](https://github.com/AnonymousPaperNo9458/Rebuttal/blob/main/TrainingDynamics/readme.md)). The results agree with the ODE prediction: over time, max p_i and variance of p_i increase, and the effective batch size decreases.
>
> ---
>
> **W3: Why does Assumption C.11 hold**
>
> We provide empirical evidence supporting Assumption C.11 in Figure 3 in the Appendix (Page 35), where the cosine similarities between $h_i$ and $E[h_i]$ are predominantly positive throughout training. Similar phenomena can be observed in other datasets at different training stages ([LINK](https://github.com/AnonymousPaperNo9458/Rebuttal/blob/main/Assumption%20C.11/readme.md)). Intuitively, when the model is far from convergence, most samples contribute to improving a shared objective, leading to gradients that point in similar directions. Hence, we believe there exist situations where Assumption C.11 holds.
>
> However, we grant that there could be situations where C.11 does not hold. In those cases, Theorem 3.6 does not hold. However, the most important result, Theorem 3.7, only requires either Theorem 3.6 or Corollary 3.5. Since Theorem 3.6 is non-local and Corollary 3.5 is local, removing Theorem 3.6 means the bound of Theorem 3.7 becomes less tight (the denominator of the RHS becomes smaller), but the main idea remains valid.
>
> ---
>
> **W4: Comparision with MetaWeightNet (MW-Net).**
>
> In Table II ([LINK](https://github.com/AnonymousPaperNo9458/Rebuttal/blob/main/Rank)), we construct two variants that match MW-Net in batch size (64) or input feature (loss), while keeping other components unchanged. Results show large batch size and feature design improve performance by 2.79% and 3.90%, respectively.
>
> ---
>
> **W5: Averaging fallacy.**
>
> First, in Table II ([LINK](https://github.com/AnonymousPaperNo9458/Rebuttal/blob/main/Rank)), our method performs best on most datasets, with only negligible gaps to the best baseline on Waterbirds (0.01%) and Texture (0.34%), indicating that the gains are not driven by any single dataset. Second, we follow the suggestion of *(Demsar JMLR 2006)* to report rank-based metrics in Table I (same link above). Our method ranks first under both criteria.
>
> ---
>
> **Q: PACS setting.** We use a *single-source single-target* setting with 12 ordered source→target pairs, training one model per pair. In Table 1, PACS results are grouped by source domain; e.g., “Art” averages Art→Photo, Art→Cartoon, and Art→Sketch.
>
> **Limitation.** We will add two limitations in the camera-ready: (1) potentially higher computational cost and environmental impact; (2) experiments on image classification only.

---

> > ### Author Rebuttal · Reviewer_Y8yx · 2026-04-03
> >
> > I would like to thank the authors for their detailed response. They have provided data / evidence that address the main concerns of my original review.

---

> > > ### Author Response · Authors · 2026-04-03
> > >
> > > Thank you very much for your thoughtful feedback and constructive suggestions. We are delighted that our rebuttal has addressed your concerns and we will incorporate the new results and discussion into the revised version. Thanks again for your time and attention.

---

### Official Review · Reviewer_XGCf · 2026-03-13

**Soundness:** 3
**Presentation:** 4
**Significance:** 3
**Originality:** 3
**Overall Recommendation:** 5
**Confidence:** 4

**Summary:**

This paper studies the practical challenges of Meta-learning for Training-data Selection (MTS), a bi-level optimization framework where a meta-network learns to weight training samples based on their contribution to validation performance. The authors identify two primary reasons why MTS often fails to outperform standard training in practice: a low Gradient Signal-to-Noise Ratio (GSNR) during optimization and the use of uninformative input features for the selection network. To address these, the authors provide a mathematical analysis demonstrating how the normalization of data weights leads to high gradient variance, particularly when weights are non-uniform. Their theoretical findings suggest that increasing the batch size can stabilize the GSNR. Additionally, they propose a richer set of input features for the meta-network—including distributional distances to class centroids and optimization dynamics like forgetting events—rather than relying solely on raw images or training loss. Experiments across four benchmarks (Waterbirds, CelebA, Texture, and PACS) show that the combined approach significantly outperforms existing data weighting baselines.

**Compliance With Llm Reviewing Policy:**

Affirmed.

**Key Questions For Authors:**

1. Could you provide a breakdown of the training time and memory consumption for your method compared to Meta-Weight-Net?

2. MTS relies heavily on the quality of the validation set to guide the meta-network. How does your method perform if the validation set is extremely small like less than 5 samples per class?

3. Why does this paper emphasis the image classification tasks? It seems that the analysis and method can be adapted into other tasks.

**Limitations:**

yes

**Strengths And Weaknesses:**

PROS:

1. Data selection is a bottleneck in modern ML, especially with the rise of synthetic data from diffusion models. Existing MTS works (like Meta-Weight-Net) focus on the network architecture rather than the optimization dynamics or feature representation. So the originality is good though MTS has been studied in these years.

2. The theoretical analysis of GSNR in the context of MTS is rigorous and provides a satisfying explanation for a common empirical observation (the instability of bi-level weighting). Lemma 3.1 and the subsequent derivations of hypergradient variance  offer a clear link between weight normalization and optimization difficulty.

3. The paper is exceptionally clear and well-organized. The transition from identifying a problem (low GSNR) to providing a theoretical cause and then an empirical solution is logical and easy to follow. The experiments are well-structured, testing the proposed method on diverse tasks involving synthetic-to-real domain transfer and group robustness. The inclusion of GSNR observation in Introduction section directly validates the theoretical claims and presents the motivation.

Concerns:

1. While the paper advocates for larger batch sizes to improve GSNR, it does not deeply explore the computational trade-offs. Bi-level optimization is already memory-intensive; requiring a batch size of 1024  might make this approach inaccessible for researchers with limited hardware, particularly for larger models.

2. The "Input Features for the Selection Network" part lacks mathematical explainations and is a little bit of short compared with another contribution.

---

> ### Author Rebuttal · Authors · 2026-03-30
>
> We thank reviewer XGCf for the diligence and thoughtful feedback. Below we provide our responses.
>
> ---
>
> **Concern1: The paper does not deeply explore the computational trade-offs. Requiring a batch size of 1024 might make this approach inaccessible for researchers with limited hardware.**
>
> We visualize the computational trade-offs in [[LINK]](https://github.com/AnonymousPaperNo9458/Rebuttal/blob/main/Computation-Performace/readme.md). As $N$ increases, our method consistently achieves higher performance at the cost of increased computation and memory. Notably, even at comparable cost ($N$=64), our method outperforms MetaWeightNet (MW-Net), indicating a better efficiency–performance trade-off.
>
> We fully agree that using a large batch may be practically inconvenient, and this is precisely why the message of this paper is important. Out of convenience, many academic researchers may choose small batch sizes when training MTS. When it performs poorly under small batch sizes, they may conclude that MTS simply doesn’t work and abandon the research direction. By drawing attention to a potential blind spot, our paper may facilitate and inspire further research on MTS.
>
> It is also worth noting that larger batch sizes do not always come with greater memory footprints. For example, we can perform gradient accumulation, which accumulates gradients from multiple small batches, to achieve similar effects. Other techniques such as mixed-precision training and gradient checkpointing may also reduce memory usage. Once again, we believe this paper will motivate more research, which may reduce memory usage further.
>
> Finally, even at moderate batch sizes like 256, our method achieves improvements of +4.42% and +3.84% over No Selection and MW-Net, respectively, while requiring only a single NVIDIA RTX A6000 GPU (same as the MW-Net).
>
> ---
>
> **Concern 2: Lack of input features mathematical explanations.**
>
> Due to space constraints, we provided the formal mathematical definitions of the input features in ​Appendix H. Input Feature Details (Pages 52-53).
>
> ---
>
> **Q1: Breakdown of the training time and memory consumption.**
>
> In our implementation, experiments are conducted on NVIDIA RTX A6000 GPUs. The computational cost is visualized in [[LINK]](https://github.com/AnonymousPaperNo9458/Rebuttal/blob/main/Computation-Performace/readme.md) and described as follows.
>
> * Number of GPUs: MW-Net requires 1 GPU, while our method requires 4 GPUs when N=1024 and can be trained on a single GPU when N=256.
> * Memory usage: Our method requires approximately 45 GB and 180 GB of GPU memory for N=256 and N=1024, respectively, compared to 17 GB for MW-Net.
> * Training time: Our method with N=1024 incurs approximately 1.9× the training time of MW-Net, while N=256 incurs approximately 1.4× overhead. For example, on the PACS Sketch→Art dataset, our method with N=1024 and N=256 requires 8.8 hours and ​6.1 hours​, respectively, compared to 4.5 hours for MW-Net.
>
> ---
>
> **Q2: How does the method perform if the validation set is extremely small?**
>
> We evaluate our method under a limited setting, where each class in the validation set contains only 5 samples. As shown below, performance drops by 3.92\% compared to the full validation setting, highlighting the importance of validation data quantity for MTS. However, even under this extremely constrained setting, our method still outperforms No Selection by +3.72% and the best baseline Aux-Clf by +2.90% on average.
>
> |Setting|Method|Waterbirds|CelebA|Texture|Avg
> |:-:|:-:|:-:|:-:|:-:|:-:|
> | |No Selection|70.31|78.52|24.44|57.76
> |5-shot|MW-Net|67.55|78.59|24.89|57.01
> |5-shot|Aux-Clf|72.78|78.77|24.20|58.58
> |5-shot|Ours|**77.51**|**81.53**|**25.41**|**61.48**
> |*Full-shot*|*Ours*|*81.81*|*85.01*|*29.38*|*65.40*
>
> ---
>
> **Q3: Why does this paper emphasize image classification tasks?**
>
> We thank the reviewer for recognizing the broader applicability of our method. Many existing papers on MTS [1-3] use image classification as the test bed, and we followed that convention.
>
> Importantly, within image classification, the datasets we use cover diverse domains and experimental settings. For example, Waterbirds and CelebA evaluate robustness to spurious correlations. Texture focuses on low-level features whereas PACS benchmarks generalization across visual styles. We contend that these datasets provide a comprehensive evaluation under varying data distributions and high/low-level visual semantics.
>
> [1] Learning to Select Pivotal Samples for Meta Re-weighting. AAAI 2023.
> [2] Meta-Weight-Net: Learning an Explicit Mapping For Sample Weighting. NeurIPS 2019.
> [3] Learning to reweight examples for robust deep learning. ICML 2018.

---

> > ### Author Rebuttal · Reviewer_XGCf · 2026-04-06
> >
> > Thank you for the thorough rebuttal. My original concerns have been fully addressed. I maintain my original positive score.

---

> > > ### Author Response · Authors · 2026-04-06
> > >
> > > Thank you very much for your thoughtful feedback and constructive suggestions. We are delighted that our rebuttal has addressed your concerns and we will incorporate the new results and discussion into the revised version. Thanks again for your time and attention.

---

### Decision · Program_Chairs · 2026-04-30

**Decision:**

Accept (spotlight)

**Comment:**

The submission considers the problem of meta-learning a network for weighting training data. This is a timely problem, given recent trends on data-driven machine learning and the high profile instances of optimising training distributions to improve model performance. There are several theoretical and practical contributions provided in the submission. It is shown how a signal to noise ratio of the gradients can impact the stability of existing approaches, a solution to this problem derived, additional meta-features are identified, and a substantial improvement in performance is achieved.

The initial reviews were already somewhat positive. The clarity and motivation of the submission were praised, and the quality of the theoretical contributions were highlighted. Some of the advice (i.e., increasing batch size) was seen as potentially not actionable in practice, and the reviewers also pointed out some ablations that would strengthen the experimental evaluation in the paper. However, many of these issues were quite minor and resolved through discussion between the authors and reviewers in the rebuttal. Given the significance and quality of the work, I strongly recommend the paper is accepted to ICML.